# Entrywise error bounds for low-rank approximations of kernel matrices

**Alexander Modell**
Department of Mathematics
Imperial College London, U.K.
`a.modell@imperial.ac.uk`

## Abstract

In this paper, we derive *entrywise* error bounds for low-rank approximations of kernel matrices obtained using the truncated eigen-decomposition (or singular value decomposition). While this approximation is well-known to be optimal with respect to the spectral and Frobenius norm error, little is known about the statistical behaviour of individual entries. Our error bounds fill this gap. A key technical innovation is a delocalisation result for the eigenvectors of the kernel matrix corresponding to small eigenvalues, which takes inspiration from the field of Random Matrix Theory. Finally, we validate our theory with an empirical study of a collection of synthetic and real-world datasets.

## 1 Introduction

Low-rank approximations of kernel matrices play a central role in many areas of machine learning. Examples include kernel principal component analysis [Schölkopf et al., 1998], spectral clustering [Ng et al., 2001, Von Luxburg, 2007] and manifold learning [Roweis and Saul, 2000, Belkin and Niyogi, 2001, Coifman and Lafon, 2006], where they serve as a core component of the algorithms, and support vector machines [Cortes and Vapnik, 1995, Fine and Scheinberg, 2001] and Gaussian process regression [Williams and Rasmussen, 1995, Ferrari-Trecate et al., 1998] where they serve to dramatically speed up computation times.

In this paper, we derive entrywise error bounds for low-rank approximations of kernel matrices obtained using the truncated eigen-decomposition (or singular value decomposition). Entrywise error bounds are important for a number of reasons. The first is practical: in applications where individual errors carry a high cost, such as system control and healthcare, we should seek methods with low entrywise error. The second is theoretical: good entrywise error bounds can lead to improved analyses of learning algorithms which use them.

For this reason, a wealth of literature has emerged establishing entrywise error bounds for a variety of matrix estimation problems, such as covariance estimation [Fan et al., 2018, Abbe et al., 2022], matrix completion [Candes and Recht, 2012, Chi et al., 2019], phase synchronisation [Zhong and Boumal, 2018, Ma et al., 2018], reinforcement learning [Stojanovic et al., 2023], community detection [Balakrishnan et al., 2011, Lyzinski et al., 2014, Eldridge et al., 2018, Lei, 2019, Mao et al., 2021] and graph inference [Cape et al., 2019, Rubin-Delanchy et al., 2022] to name a few.

### 1.1 Contributions

- Our main result (Theorem 1) is an entrywise error bound for the low-rank approximation of a kernel matrix. Under regularity conditions, we find that for kernels with polynomial eigenvalue decay, $\lambda_i = \mathcal{O}(i^{-\alpha})$, we require a polynomial-rank approximation, $d = \Omega(n^{1/\alpha})$,

38th Conference on Neural Information Processing Systems (NeurIPS 2024).

to achieve entrywise consistency. For kernels with exponential eigenvalue decay, $\lambda_i = \mathcal{O}(e^{\beta i^\gamma})$, we require a (poly)log-rank approximation, $d > \log^{1/\gamma}(n^{1/\beta})$.

- The main technical contribution of this paper is to establish a delocalisation result for the eigenvectors of the kernel matrix corresponding to small eigenvalues (Theorem 3), the proof of which draws on ideas from the Random Matrix Theory literature. To our knowledge, this is the first such result for a random matrix with non-zero mean and dependent entries.

- Along the way, we prove a novel concentration inequality for the distance between a random vector (with a potentially non-zero mean) and a subspace (Lemma 1), which may be of independent interest.

- We complement our theory with an empirical study on the entrywise errors of low-rank approximations of the kernel matrices on a collection of synthetic and real datasets.

## 1.2   Related work

Some complementary results to ours use the Johnson-Lindenstrauss lemma [Johnson and Lindenstrauss, 1982] to bound the entrywise error of low-rank matrix approximations obtained via random projections [Srebro and Shraibman, 2005, Alon et al., 2013, Udell and Townsend, 2019, Budzinskiy, 2024a,b]. In Section 3.2 we discuss these results in more detail and compare them with ours.

Our proof strategy draws heavily on ideas from the Random Matrix Theory literature, where delocalisation results have been established for certain classes of zero-mean random matrices with independent entries [Erdős et al., 2009b,a, Tao and Vu, 2011, Rudelson and Vershynin, 2015, Vu and Wang, 2015]. In addition, our result is made possible by recent *relative* eigenvalue concentration bounds for kernel matrices [Braun, 2006, Valdivia, 2018, Barzilai and Shamir, 2023], which improve upon classical *absolute* concentration bounds [Rosasco et al., 2010] which would not provide sufficient control for our purposes.

## 1.3   Big-$\mathcal{O}$ notation and frequent events

We use the standard big-$\mathcal{O}$ notation where $a_n = \mathcal{O}(b_n)$ (resp. $a_n = \Omega(b_n)$) means that for sufficiently large $n$, $a_n \leq C b_n$ (resp. $a_n \geq C b_n$) for some constant $C$ which doesn't depend on the parameters of the problem. We will occasionally write $a_n \lesssim b_n$ to mean that $a_n = \mathcal{O}(b_n)$.

In addition, we say that an event $E_n$ holds *with overwhelming probability* if for *every* $c > 0$, $\mathbb{P}(E_n) \geq 1 - \mathcal{O}(n^{-c})$, where the hidden constant is allowed to depend on $c$.

## 2   Setup

We begin by describing the setup of our problem. We suppose that we observe $n$, $p$-dimensional data points $\{x_i\}_{i=1}^n$, which we assume were drawn i.i.d. from some probability distribution $\rho$, supported on a set $\mathcal{X}$. Given a symmetric kernel $k : \mathcal{X} \times \mathcal{X} \to \mathbb{R}$, we construct the $n \times n$ kernel matrix $K$, with entries

$$K(i,j) := k(x_i, x_j).$$

We will assume throughout that the kernel is positive-definite, continuous and bounded. Let $\widehat{K}_d$ denote the "best" rank-$d$ approximation of $K$, in the sense that $\widehat{K}_d$ satisfies

$$\widehat{K}_d := \underset{K':\mathrm{rank}(K')=d}{\arg\min} \|K - K'\|_\xi,  \tag{1}$$

where $\|\cdot\|_\xi$ is a rotation-invariant norm, such as the spectral or Frobenius norm. Then by the Eckart-Young-Mirsky theorem [Eckart and Young, 1936, Mirsky, 1960], $\widehat{K}_d$ is given by the truncated eigen-decomposition of $K$, i.e.

$$\widehat{K}_d = \sum_{i=1}^d \widehat{\lambda}_i \widehat{u}_i \widehat{u}_i^\top$$

where $\{\widehat{\lambda}_i\}_{i=1}^n$ are the eigenvalues of $K$ (counting multiplicities) in decreasing order, and $\{\widehat{u}_i\}_{i=1}^n$ are corresponding eigenvectors.

We now introduce some population quantities which will form the basis of our theory. Let $L^2_\rho$ denote the Hilbert space of real-valued square integrable functions with respect to $\rho$, with the inner product defined as $\langle f, g \rangle_\rho = \int f(x)g(x)\mathrm{d}\rho(x)$. We define the integral operator $\mathcal{K} : L^2_\rho \to L^2_\rho$ by

$$(\mathcal{K}f)(x) = \int k(x,y)f(y)\mathrm{d}\rho(y)$$

which is the infinite sample limit of $\frac{1}{n}K$. The operator $\mathcal{K}$ is self-adjoint and compact [Hirsch and Lacombe, 1999], so by the spectral theorem for compact operators, there exists a sequence of eigenvalues $\{\lambda_i\}_{i=1}^\infty$ (counting multiplicities) in decreasing order, with corresponding eigenfunctions $\{u_i\}_{i=1}^\infty$ which are orthonormal in $L^2_\rho$, such that

$$\mathcal{K}u_i = \lambda_i u_i.$$

Moreover, by the classical Mercer's theorem [Mercer, 1909, Steinwart and Scovel, 2012], the kernel $k$ can be decomposed into

$$k(x,y) = \sum_{i=1}^\infty \lambda_i u_i(x) u_i(y) \tag{2}$$

where the series converges absolutely and uniformly in $x, y$.

## 3 Entrywise error bounds

This section is devoted to our main theoretical result. We begin by discussing our assumptions, before presenting our main theorem and giving some special cases of kernels which fit within our framework.

In our asymptotics, we will assume that $k$ and $\rho$ are fixed, and that the number of samples $n$ goes to infinity. This places us in the so-called low-dimensional regime, in which the dimension $p$ of the input space is considered fixed.

We shall assume that the eigenvalues of the kernel exhibit either polynomial decay, i.e. $\lambda_i = \mathcal{O}(i^{-\alpha})$ for some $\alpha > 1$, or (nearly) exponential decay, i.e. $\lambda_i = \mathcal{O}(e^{-\beta i^\gamma})$ for some $\beta > 0$ and $0 < \gamma \le 1$. We will refer to these two hypotheses as (P) and (E) respectively. We also assume a corresponding hypothesis on the supremum-norm growth of the eigenfunctions. Under (P), we assume that $\|u_i\|_\infty = \mathcal{O}(i^r)$ with $\alpha > 2r + 1$, and under (E), we assume that $\|u_i\|_\infty = O(e^{si^\gamma})$ with $\beta > 2s$.

Our eigenvalue decay hypothesis is commonplace in the kernel literature [Braun, 2006, Ostrovskii and Rudi, 2019, Xu, 2018, Lei, 2021], and can be related to the smoothness of the kernel. For example, the decay of the eigenvalues is directly implied by a Hölder or Sobolev-type smoothness hypothesis on the kernel (see, for example, Nicaise [2000], Belkin [2018], Section 2.2 of Xu [2018], Section 5 of Valdivia [2018], Scetbon and Harchaoui [2021] and Proposition 2 in this paper). We don't consider a finite-rank (say, $D$) hypothesis, since in this case the maximum entrywise error is trivially zero whenever $d \ge D$.

Our hypothesis on the supremum norm of the eigenfunctions is necessary to control the deviation of the sample eigenvectors from their corresponding population eigenfunctions, and is a requirement of eigenvalue bounds we employ. In the literature, it is common to see much stronger assumptions, such as uniformly bounded eigenfunctions [Williamson et al., 2001, Lafferty et al., 2005, Braun, 2006], which do not hold for many commonly-used kernels (see Mendelson and Neeman [2010], Steinwart and Scovel [2012], Zhou [2002] and Barzilai and Shamir [2023] for discussion). This assumption is reminiscent of the *incoherence* assumption [Candes and Recht, 2012] in the low-rank matrix estimation literature — a supremum norm bound on population eigenvectors — which governs the hardness of many compressed sensing and eigenvector estimation problems [Candès and Tao, 2010, Keshavan et al., 2010, Chi et al., 2019, Abbe et al., 2020, Chen et al., 2021].

In addition, we introduce a regularity hypothesis, which we will refer to as (R), which relates to the following two quantities:

$$\Delta_i = \max_{j \ge i} \{\lambda_j - \lambda_{j+1}\}$$

which measures the largest eigengap after a certain point in the spectrum, and

$$\Gamma_i = \sum_{j=i+1}^\infty \left( \int u_j(x)\mathrm{d}\rho(x) \right)^2$$

| | $\lambda_i$ | $\|u_i\|_\infty$ | | | $\Delta_i$ | $\Gamma_i$ | |
|---|---|---|---|---|---|---|---|
| (P) | $\mathcal{O}\left(i^{-\alpha}\right)$ | $\mathcal{O}\left(i^{r}\right)$ | $\alpha > 2r+1$ | (R) | $\mathcal{O}(\lambda_i^a)$ | $\mathcal{O}(\lambda_i^b)$ | $1 \le a < b/16$ |
| (E) | $\mathcal{O}\left(e^{-\beta i^\gamma}\right)$ | $\mathcal{O}\left(e^{si^\gamma}\right)$ | $\beta > 2s, 0 < \gamma \le 1$ | | | | |

Table 1: Summary of the hypotheses (P), (E) and (R).

which measures the squared residual after projecting the unit-norm constant function onto the first $i$ eigenfunctions. Under (R), we assume that $\Delta_i = \Omega\left(\lambda_i^a\right)$ and $\Gamma_i = \mathcal{O}\left(\lambda_i^b\right)$ with $1 \le a < b/16 \le \infty$.

A sufficient condition for (R) to hold, is that the first eigenfunction is constant. This holds, for example, when $k$ is a dot-product kernel and $\rho$ is a uniform distribution on a hypersphere. In such scenarios, $\Gamma_i = 0$ for all $i \ge 1$ and it is not necessary to make any assumptions on the eigengap quantity $\Delta_i$. We remark that (R) permits repeated eigenvalues in the spectrum of $\mathcal{K}$, which occur for many commonly-used kernels, but which are often precluded in the literature [Hall and Horowitz, 2007, Meister, 2011, Lei, 2014, 2021].

The hypotheses (P), (E) and (R) are summarised in Table 1. We are now ready to state our main theorem.

**Theorem 1.** *Suppose that $k$ is a symmetric, positive-definite, continuous and bounded kernel and $\rho$ is a probability measure which satisfy (R) and one of either (P) or (E). If the hypothesis (P) holds and $d = \Omega\left(n^{1/\alpha}\right)$, then*

$$\left\|\widehat{K}_d - K\right\|_{\max} = \mathcal{O}\left(n^{-\frac{\alpha-1}{\alpha}}\log(n)\right) \tag{3}$$

*with overwhelming probability. If the hypothesis (E) holds and $d > \log^{1/\gamma}(n^{1/\beta})$, then*

$$\left\|\widehat{K}_d - K\right\|_{\max} = \mathcal{O}\left(n^{-1}\right)$$

*with overwhelming probability.*

### 3.1 Special cases

In this section, we provide some examples of kernels which satisfy the assumptions of Theorem 1. Proofs of the propositions in this section are given in Section A of the appendix. We start with a canonical example of a radial basis kernel.

**Proposition 1.** *Suppose $k(x, y) = \exp\left(-\|x-y\|^2/2\omega^2\right)$ is a radial basis kernel, and $\rho \sim \mathcal{N}(0, \sigma^2 I_p)$ is a isotropic Gaussian distribution on $\mathbb{R}^p$. Then the hypotheses (E) and (R) are satisfied with*

$$\beta = \log\left(\frac{1+\upsilon+\sqrt{1+2\upsilon}}{\upsilon}\right), \qquad \gamma = 1$$

*where $\upsilon := 2\sigma^2/\omega^2$.*

For this example, the eigenvalues and eigenfunctions were explicitly calculated in Zhu et al. [1997] (see also Shi et al. [2008] and Shi et al. [2009]), and we are able to verify the assumptions by direct calculation.

For our second example, we consider the case that $\rho$ is the uniform distribution on a hypersphere $\mathbb{S}^{p-1}$, and $k$ is a dot-product kernel. In this setting, we are able to replace our assumptions with a smoothness hypothesis on the kernel. Note that this class of kernels includes those which are functions of Euclidean distance, since on the sphere we have the identity $\|x-y\|^2 = 2 - 2\langle x, y\rangle$.

**Proposition 2.** *Suppose that*

$$k(x, y) = f(\langle x, y\rangle) \equiv \sum_{i=0}^{\infty} b_i\left(\langle x, y\rangle\right)^i$$

*is a dot-product kernel and $\rho$ is the uniform distribution on the hypersphere $\mathbb{S}^{p-1}$ with $p \ge 3$. If there exists $a > (p^2 - 4p + 5)/2$ such that $b_i = \mathcal{O}(i^{-a})$, then (P) and (R) are satisfied with*

$$\alpha = \frac{2a + p - 3}{p - 2}.$$

*Alternatively, if there exists $0 < r < 1$ such that $b_i = \mathcal{O}(r^i)$, then (E) and (R) are satisfied with*

$$\beta = \frac{(p-1)!}{C} \log(1/r), \qquad \gamma = \frac{1}{p-1}$$

*for some universal constant $C > 0$.*

In this example, the eigenfunctions posses the property that they do not depend on the choice of kernel, and are made up of *spherical harmonics* [Smola et al., 2000]. In particular, the first eigenfunction is constant, and therefore (R) is satisfied automatically. The eigenvalue bounds are derived in Scetbon and Harchaoui [2021], and we make use of a supremum norm bound for spherical harmonics in Minh et al. [2006].

## 3.2 Comparison with random projections and the Johnson-Lindenstrauss lemma

We pause here to consider how our entrywise bounds compare with existing bounds in the literature for low-rank matrix obtained via random projections [Srebro and Shraibman, 2005, Alon et al., 2013, Udell and Townsend, 2019, Budzinskiy, 2024a,b].

For an $n \times n$ symmetric, positive semi-definite matrix $M$ with bounded entries, the Johnson-Lindenstrauss lemma [Johnson and Lindenstrauss, 1982] can be used to show the existence of a rank-$d$ approximation $\widehat{M}_d$ whose entrywise error is bounded by $\varepsilon$ when $d = \Omega(\varepsilon^{-2} \log(n))$.

The proof is via a probabilistic construction. Let $X$ be an $n \times n$ matrix such that $M = XX^\top$, and for some $d \leq n$, let $R$ be an $n \times d$ matrix with i.i.d. entries from $\mathcal{N}(0, 1/d)$. Then, the randomised low-rank approximation

$$\widehat{M}_d := XRR^\top X^\top \tag{4}$$

achieves the desired bound with high probability. Here, we state a adaptation of Theorem 1.4 of Alon et al. [2013] which makes the probabilistic construction from the proof explicit.

**Theorem 2.** *Let $M$ be an $n \times n$ positive semi-definite matrix with bounded entries, and $\widehat{M}_d$ be a randomised rank-$d$ approximation of $M$ described in* (4)*. Then*

$$\left\| \widehat{M}_d - M \right\|_{\max} = \mathcal{O}\left( \sqrt{\frac{\log(n)}{d}} \right)$$

*with overwhelming probability.*

To obtain a polynomial entrywise error rate, i.e. $\mathcal{O}(n^{-c})$ for some $c > 0$, with Theorem 2, requires the rank $d$ to be polynomial in $n$. In contrast, under our hypothesis (E), we are able to obtain a polynomial entrywise error rate using a spectral low-rank approximation with only poly-logarithmic rank. In addition, while our entrywise error bounds are $o(n^{-1/2})$ for the cases we consider, this rate can never be achieved, regardless of the choice of rank $d$, by (4) with Theorem 2.

On the flip side, Theorem 2 holds for arbitrary positive semi-definite matrix with bounded entries, whereas our theorem only holds for kernel matrices satisfying the hypotheses in Table 1.

## 4 Proof of Theorem 1

In this section, we outline the proof of Theorem 1. Without loss of generality, we will assume that $k$ is upper bounded by one. We cover the main details here, and defer some of the technical details to the appendix. By the Eckart-Young-Mirsky theorem, we have that

$$\left\| \widehat{K}_d - K \right\|_{\max} := \max_{1 \leq i,j \leq n} \left| \widehat{K}_d(i,j) - K(i,j) \right| = \left| \max_{1 \leq i,j \leq n} \sum_{l=d+1}^{n} \widehat{\lambda}_l \widehat{u}_l(i) \widehat{u}_l(j) \right|$$

$$\leq \sum_{l=d+1}^{n} \left| \widehat{\lambda}_l \right| \cdot \max_{d < l \leq n} \| \widehat{u}_l \|_\infty^2 .$$

Using a concentration bound due to Valdivia [2018], we are able to show that

$$\sum_{l=d+1}^{n} \left| \widehat{\lambda}_l \right| = \begin{cases} \mathcal{O}\left( n^{1/\alpha} \log(n) \right) & \text{under (P) with } d = \Omega\left( n^{1/\alpha} \right) \\ \mathcal{O}(1) & \text{under (E) with } d > \log^{1/\gamma}(n^{1/\beta}). \end{cases} \tag{5}$$

with overwhelming probability, the details of which are given in Section B of the appendix. Then, the proof boils down to showing the following result, which we state as an independent theorem.

**Theorem 3.** *Assume the setting of Theorem 1, then simultaneously for all $d + 1 \leq l \leq n$,*

$$\|\widehat{u}_l\|_\infty = \mathcal{O}\left(n^{-1/2}\right) \tag{6}$$

*with overwhelming probability.*

When a unit eigenvector satisfies (6) (up to log factors), it is said to be *completely delocalised*. There is a now expansive literature in the field of Random Matrix Theory proving the delocalisation of the eigenvectors of certain mean-zero random matrices with independent entries [Erdős et al., 2009b,a, Tao and Vu, 2011, Rudelson and Vershynin, 2015, Vu and Wang, 2015]. Theorem 3 may be of independent interest since to our knowledge, it is the first eigenvector delocalisation result for a random matrix with non-zero mean and dependent entries.

To prove Theorem 3, we take inspiration from a proof strategy employed in Tao and Vu [2011] (see also Erdős et al. [2009b]) which makes use of an identity relating the eigenvalues and eigenvectors of a matrix with that of its principal minor. The non-zero mean, and dependence between the entries of a kernel matrix present new challenges which require novel technical insights and tools and make up the bulk of our technical contribution.

*Proof of Theorem 3.* By symmetry and a union bound, to prove Theorem 3, it suffices to establish the bound for the first coordinate of $\widehat{u}_l$ for some an arbitrary index $d < l \leq n$. We shall let $\widetilde{K}$ denote the bottom right principal minor of $K$, that is the $n - 1 \times n - 1$ matrix such that

$$K = \begin{pmatrix} z & y^\top \\ y & \widetilde{K} \end{pmatrix}$$

where $z = k(x_1, x_1)$ and $y = (k(x_1, x_2), \ldots, k(x_1, x_n))^\top$. We will denote the ordered eigenvalues and corresponding eigenvectors of $\widetilde{K}$ by $\{\widetilde{\lambda}_l\}_{l=1}^{n-1}$ and $\{\widetilde{u}_l\}_{l=1}^{n-1}$ respectively. By Lemma 41 of Tao and Vu [2011], we have the following remarkable identity:

$$\widehat{u}_l(1)^2 = \frac{1}{1 + \sum_{j=1}^{n-1}\left(\widetilde{\lambda}_j - \widehat{\lambda}_i\right)^{-2}\left(\widetilde{u}_j^\top y\right)^2}. \tag{7}$$

In addition, Cauchy's interlacing theorem tells us that the eigenvalues of $K$ and $\widetilde{K}$ interlace, i.e. $\widehat{\lambda}_i \leq \widetilde{\lambda}_i \leq \widehat{\lambda}_{i+1}$ for all $1 \leq i \leq n - 1$. By (5) we have that $|\widehat{\lambda}_i| = \mathcal{O}(1)$ for all $d + 1 \leq i \leq n$ with overwhelming probability and so by Cauchy's interlacing theorem, we can find a set of indices $J \subset \{d + 1, \ldots, n - 1\}$ with $|J| \geq (n - d)/2$ such that $|\widetilde{\lambda}_j - \widehat{\lambda}_i| = \mathcal{O}(1)$ for all $j \in J$. Combining this observation with (7), we have that

$$\sum_{j=1}^{n-1}\left(\widetilde{\lambda}_j - \widehat{\lambda}_i\right)^{-2}\left(\widetilde{u}_j^\top y\right)^2 \geq \sum_{j \in J}\left(\widetilde{\lambda}_j - \widehat{\lambda}_i\right)^{-2}\left(\widetilde{u}_j^\top y\right)^2 \gtrsim \|\pi_J(y)\|^2. \tag{8}$$

where $\pi_J$ denotes the orthogonal projection onto the subspace spanned by $\{\widetilde{u}_j\}_{j \in J}$. So, to establish (6), it suffices to show that

$$\|\pi_J(y)\|^2 = \Omega(n) \tag{9}$$

with overwhelming probability. We condition on $x_1$, so that $y$ is a vector of independent random variables and denote its conditional mean by $\bar{y}$, which is a constant vector whose entries are less than one. In addition, each entry of $y$ has common conditional variance which we denote by $\sigma^2 = \mathbb{E}_{x \sim F}\{k^2(x_1, x)\}$.

To obtain the lower bound (9), we prove a novel concentration inequality for the distance between a random vector and a subspace, which may be of independent interest. Our lemma generalises a similar result in Tao and Vu [2011, Lemma 43] which holds only for random vectors with zero mean and unit variance. The proof is provided in Section C of the appendix.

**Lemma 1.** *Let $y \in \mathbb{R}^n$ be a random vector with mean $\bar{y} := \mathbb{E}y$ whose entries are independent, have common variance $\sigma^2$ and are bounded in $[0,1]$ almost surely. Let $H$ be a subspace of dimension $q \geq 64/\sigma^2$ and $\pi_H$ the orthogonal projection onto $H$. If $H$ is such that $\|\pi_H(\bar{y})\| \leq 2(\sigma^2 q)^{1/4}$, then for any $t \geq 8$*

$$\mathbb{P}\left(\left|\|\pi_H(y)\| - \sigma q^{1/2}\right| \geq t\right) \leq 4\exp\left(-t^2/32\right).$$

*In particular, one has*

$$\|\pi_H(y)\| = \sigma q^{1/2} + \mathcal{O}\left(\log^{1/2}(n)\right) \tag{10}$$

*with overwhelming probability.*

Returning to the main thread, we claim for the moment that $\|\pi_J(\bar{y})\| \leq 2|J|^{1/4}$. Then, by Lemma 1 we have that, conditional on $x_1$,

$$\|\pi_J(y)\|^2 \gtrsim |J| \geq (n-d)/2 = \Omega(n)$$

with overwhelming probability. This holds for all $x_1 \in \mathcal{X}$ and therefore establishes (9). To complete the proof, then, it remains to prove our claim, and it is here where we require the regularity hypothesis (R). The proof of the claim is quite involved, so we defer the details to Section D of the appendix, given which, the proof of Theorem 3 is complete.

$\square$

## 5 Experiments

### 5.1 Datasets and setup

To see how our theory translates into practice, we examine the maximum entrywise error of the low-rank approximations of kernel matrices derived from a synthetic dataset and a collection of five real-world data sets, which are summarised in the following table[1].

| Dataset | Description | # Instances | # Dimensions |
|---------|-------------|-------------|--------------|
| GMM | simulated data from a Gaussian mixture model | 1,000 | 10 |
| Abalone | physical measurements of Abalone trees | 4,177 | 7 |
| Wine Quality | physicochemical measurements of wines | 6,497 | 11 |
| MNIST | handwritten digits | 10,000 | 784 |
| 20 Newsgroups | tf-idf vectors from newsgroup messages | 11,314 | 21,108 |
| Zebrafish | gene expression in zebrafish embryo cells | 6,346 | 5,434 |

Additional details about the dataset are provided in Section E of the appendix.

For the purpose of our experiment, we employ kernels in the class of *Matérn kernels*, of the form

$$k_\nu(x,y) = \frac{2^{1-\nu}}{\Gamma(\nu)}\left(\sqrt{2\nu}\frac{\|x-y\|}{\omega}\right)^\nu K_\nu\left(\sqrt{2\nu}\frac{\|x-y\|}{\omega}\right)$$

where $\Gamma$ denotes the gamma function, and $K_\nu$ is the modified Bessel function of the second kind. The class of Matérn kernels is a generalisation of the radial basis kernel, with an additional parameter $\nu$ which governs the smoothness of the resulting kernel. When $\nu = 1/2$, we obtain the non-differentiable exponential kernel, and in the $\nu \to \infty$ limit, we obtain the infinitely-differentiable radial basis kernel. For the intermediate values $\nu = 3/2$ and $\nu = 5/2$, we obtain, respectively, once and twice-differentiable functions.

The optimal choice of the bandwidth parameter is problem-dependent, and in supervised settings is typically chosen using cross-validation. In unsupervised settings, it is necessary to rely on heuristics,

---

[1]Code to reproduce the experiments in this section can be found at
https://gist.github.com/alexandermodell/b16b0b29b6d0a340a23dab79219133f2.

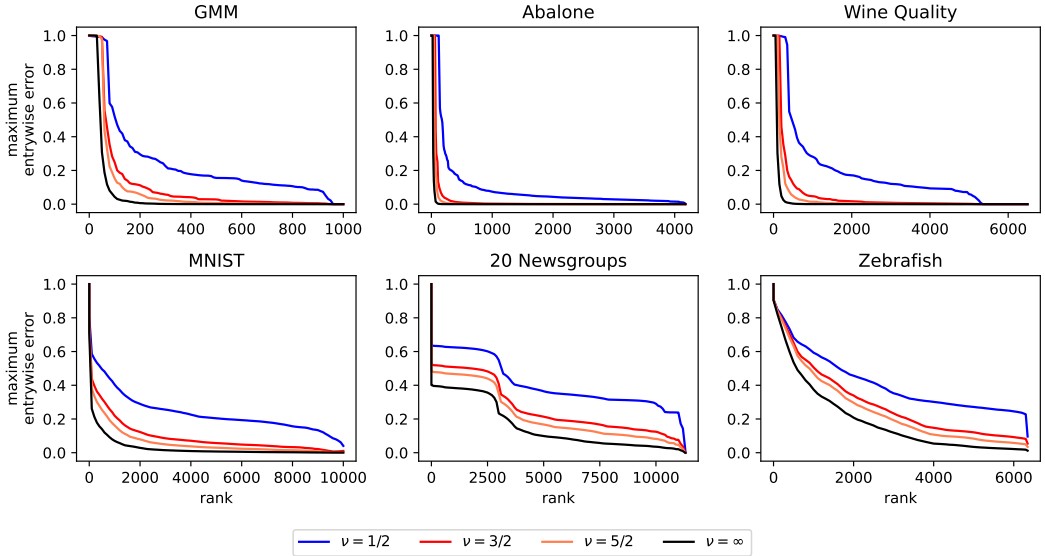

Figure 1: The maximum entrywise error against rank for low-rank approximations of kernel matrices constructed from a collection of datasets. The kernel matrices are constructed using Matérn kernels with a range of smoothness parameters, each of which is represented by a line in each plot. Details of the experiment are provided in Section 5.

and for this experiment, we use the popular *median heuristic* [Flaxman et al., 2016, Mooij et al., 2016, Mu et al., 2016, Garreau et al., 2017], which has been shown to perform well in practice.

For each dataset, we construct four kernel matrices using Matérn kernels with smoothness parameters $\nu = \frac{1}{2}, \frac{3}{2}, \frac{5}{2}, \infty$, each time selecting the bandwidth using the median heuristic. For each kernel, we compute the best rank-$d$ low-rank approximation of the kernel matrix using the `svds` function in the SciPy library for Python [Virtanen et al., 2020]. We do this for a range of ranks $d$ from 1 to $n$, where $n$ is the number of instances in the dataset, and record the entrywise errors.

## 5.2 Interpretation of the results

Figure 1 shows the maximum entrywise errors for each dataset and kernel. For comparison, the Frobenius norm errors are plotted in Figure 2 in Section E of the appendix.

As predicted by our theory, for the four "low-dimensional" datasets, *GMM*, *Abalone*, *Wine Quality* and *MNIST*, the maximum entrywise decays rapidly as we increase the rank of the approximation, with the exception of the highly non-smooth $v = \frac{1}{2}$ kernel, for which the maximum entrywise error decays much more slowly. In addition, the decay rates of the maximum entrywise error are in order of the smoothness of the kernels.

For the "high-dimensional" datasets, *20 Newsgroups* and *Zebrafish*, a different story emerges. Even for the smooth radial basis kernel ($\nu = \infty$), the maximum entrywise error decays very slowly. This would suggests that our theory does potentially *not* carry over to the high-dimensional setting, and that caution should be taken when employing low-rank approximations for such data. Interestingly, the *20 Newsgroups* dataset exhibits a sharp drop in maximum entrywise error between $d = 2500$ and $d = 3000$ which *is not* seen in the decay of the Frobenius norm error (Figure 2 in Section E).

## 6 Limitations and open problems

To conclude, we discuss some of the limitations of our theory, as well as some of the open problems.

## 6.1 Limitations of our theory

*Positive semi-definite kernels.* One significant limitation of our theory is the assumption that the kernel is positive semi-definite and continuous. This condition is known as Mercer's condition in the literature and ensures that the spectral decomposition of the kernel (2) converges uniformly, however we don't actually require such a strong notion of convergence for our theory. Valdivia [2018, Lemma 22] show that the decomposition converges *almost surely* under a much weaker condition which is implied by our hypotheses (P) and (E). The only other places we need this assumption is to make use of results in Rosasco et al. [2010] and Tang et al. [2013]. These results make heavy use of reproducing kernel Hilbert space technology though it seems plausible that they could be generalised to the indefinite setting using the framework of Krein spaces [Ong et al., 2004, Lei, 2021].

*Low-dimensional setting.* In our asymptotics, we explicitly assume that the dimension of the input space remains fixed as the number of sample increases, which places us in the so-called low-dimensional setting. We do not consider the high-dimensional setting, however our empirical experiments suggest that our conclusions may not carry over.

*Verification of the assumptions.* While there is a established literature studying the eigenvalue decay of kernels under general probability measures [Kühn, 1987, Cobos and Kühn, 1990, Ferreira and Menegatto, 2013, Belkin, 2018, Li et al., 2024], except in very specialised settings (such as Propositions 1 and 2), control of the eigenfunctions is typically out of reach. This makes verifying the assumptions of our theory under general probability distributions quite challenging. This is a widespread limitation of many theoretical analyses in the kernel literature, and for an extended discussion, we refer the reader to Barzilai and Shamir [2023].

## 6.2 Open problems

*Randomised low-rank approximations.* While the truncated spectral decomposition provides the "ideal" low-rank approximation, it requires computing the whole kernel matrix which can be prohibitive for very large datasets. Randomised low-rank approximations, such as the *randomised SVD* [Halko et al., 2011], the *Nyström* method [Williams and Seeger, 2000, Drineas et al., 2005] and *random Fourier features* [Rahimi and Recht, 2007, 2008], have emerged as efficient alternatives, and there is an extensive body of literature examining their statistical performance [Drineas et al., 2005, Rahimi and Recht, 2007, Belabbas and Wolfe, 2009, Boutsidis et al., 2009, Kumar et al., 2009a,b, Gittens, 2011, Gittens and Mahoney, 2013, Altschuler et al., 2016, Derezinski et al., 2020]. However, their primary focus is on classical error metrics such as the spectral and Frobenius norm errors and an entrywise analysis would presumably provide greater insights into these approximations, particularly given recently observed multiple-descent phenomena [Derezinski et al., 2020].

*Lower bounds.* At present, it is unclear whether the bounds we obtain are tight, or indeed whether the truncated spectral decomposition itself is optimal with respect the the entrywise error. An interesting direction for future research would be to investigate lower bounds to understand the fundamental limits of this problem.

# Acknowledgements

The author thanks Nick Whiteley, Yanbo Tang and Mahmoud Khabou for helpful discussions and Annie Gray for providing code to preprocess the *20 Newsgroups* and *Zebrafish* datasets.

This work was supported by the Engineering and Physical Sciences Research Council [grant EP/X002195/1].

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

# Appendix

## A  Proof of Propositions 1 and 2

For notational simplicity, in this section we will assume in this section that the eigenvalues and eigenfunctions are indexed from 0 rather than 1.

### A.1  Proof of Proposition 1

We will begin by reducing the problem of verifying our assumptions under $\rho$ to verifying them under the probability measure associated with the univariate Gaussian distribution $\mathcal{N}(0, \sigma^2)$, which we will denote by $\mu$.

Let $\underline{\mathcal{K}} : L_\rho^2 \to L_\rho^2$ denote the integral operator associated with the kernel $k$ and the measure $\rho$, and let $\{\underline{\lambda}_i\}$ denote its eigenvalues, arranged in descending order, and $\{\underline{u}_i\}$ denote their corresponding eigenfunctions. By the rotation invariance of both $k$ and $\rho$, the operator $\underline{\mathcal{K}}$ may be written as the $p$-fold tensor product

$$\underline{\mathcal{K}} = \mathcal{K} \otimes \cdots \otimes \mathcal{K}$$

where $\mathcal{K} : L_\mu^2 \to L_\mu^2$ denotes the integral operator associated with the kernel $k$ and the univariate Gaussian measure $\mu$. Let $\{\lambda_i\}$ denote its eigenvalues, arranged in descending order, and $\{u_i\}$ denote their corresponding eigenfunctions. Then, the eigenvalues and eigenfunctions of $\underline{\mathcal{K}}$ and $\mathcal{K}$ are related in the following way (see Shi et al. [2008] or Fasshauer and McCourt [2012]). For every $i$, there exists $i_1, \ldots, i_p$ satisfying $\sum_{j=1}^p i_j = i$ such that

$$\underline{\lambda}_i = \prod_{j=1}^p \lambda_{i_j} \qquad \text{and} \qquad \underline{u}_i(x) = \prod_{j=1}^p u_{i_j}(x^j) \tag{11}$$

for all $x = (x^1, \ldots, x^p)^\top \in \mathbb{R}^p$. Now suppose that $\lambda_i = \Theta\left(e^{-\beta i}\right)$, then

$$\underline{\lambda}_i = \prod_{j=1}^p \lambda_{i_j} \asymp \prod_{j=1}^p e^{-\beta i_j} = e^{-\beta \sum_j i_j} = e^{-\beta i},$$

and suppose that $\|u_i\|_\infty = \mathcal{O}(e^{si})$ for some $s < \beta/2$. Then

$$\|\underline{u}_i\|_\infty \le \prod_{j=1}^p \|u_{i_j}\|_\infty \lesssim \prod_{j=1}^p e^{si_j} = e^{si}$$

Therefore to prove that (E) hold under $\rho$, it suffices to show that it holds under $\mu$.

Shi et al. [2008] provide an explicit formula for the eigenvalues and eigenfunctions of $\mathcal{K}$, which is a refinement of an earlier result of Zhu et al. [1997].

Let $\upsilon := 2\sigma^2/\omega^2$ and let $H_i(x)$ be the $i$th order Hermite polynomial. Then the eigenvalues and eigenfunctions of $\mathcal{K}$ are given by

$$\lambda_i = \sqrt{\frac{2}{1 + \upsilon + \sqrt{1 + 2\upsilon}}} \left(\frac{\upsilon}{1 + \upsilon + \sqrt{1 + 2\upsilon}}\right)^i$$

$$u_i(x) = \frac{(1 + 2\beta)^{1/8}}{\sqrt{2^i i!}} \exp\left(-\frac{x^2}{2\sigma^2} \frac{\sqrt{1 + 2\beta} - 1}{2}\right) H_i\left(\left(\frac{1}{4} + \frac{\beta}{2}\right)^{1/4} \frac{x}{\sigma}\right).$$

Therefore, we have $\lambda_i = C_1 e^{-\beta i}$ where

$$\beta = \log\left(\frac{1 + \upsilon + \sqrt{1 + 2\upsilon}}{\upsilon}\right).$$

We will now show that each $u_i$ is uniformly bounded. By a change of variables, we can write $u_i$ as

$$u_i(x) = \frac{C_2}{\sqrt{2^i i!}} e^{-y^2} H_i(y)$$

for some $y \in \mathbb{R}$. On the other hand, we have the following inequality due to Indritz [1961]. For all $x \in \mathbb{R}$,

$$e^{-x^2} H_i(x) \leq 1.09\sqrt{2^i i!}.$$

Therefore $u_i(x) \leq 1.09 C_2$ for all $x$. We can use the fact that $H_i(x)$ is either odd or even to obtain an analogous lower bound. Therefore $\|u_i\|_\infty \leq 1.09 C_2$ for all $i$, so $s = 0$ and (E) holds.

We will now show that $\Gamma_i = 0$ for all $i \geq 1$ so that (R) holds with $b = +\infty$ and there is no requirement on the eigengaps $\Delta_i$.

Expanding $\int u_i(x)\mathrm{d}\mu(x)$, collecting exponential terms and applying a change-of-variables, one can calculate that

$$\int u_i(x)\mathrm{d}\mu(x) = C_3 \int_{-\infty}^{+\infty} e^{-y^2} H_i(y)\mathrm{d}y.$$

It is a standard result that $e^{-y^2} H_i(y) = 0$ as long as $i \neq 0$ [Gradshteyn and Ryzhik, 2014], and therefore $\left|\int u_i(x)\mathrm{d}\mu(x)\right| = 0$ for all $i \geq 1$, and by (11), we have that $\Gamma_i = 0$ for all $i \geq 1$.

### A.2   Proof of Proposition 2

For any $p \geq 3$, dot-product kernels with respect to the uniform measure on the sphere exhibit the spectral decomposition

$$k(x,y) = \sum_{l=0}^{\infty} \lambda_l^* \sum_{m=1}^{N_l} u_{l,m}^*(x) u_{l,m}^*(y)$$

where the eigenfunctions $\{u_{l,m}^*\}$ are the $m$th spherical harmonic of degree $l$, $N_l = \frac{2l+p-2}{l}\binom{l+p-3}{p-2} = \mathcal{O}(l^{p-2})$ is the number of harmonics of each degree, and $\{\lambda_l^*\}$ are the distinct eigenvalues [Smola et al., 2000].

The first spherical harmonic is a constant function, and therefore by the orthogonality of the eigenfunctions in $L_\rho^2$, $\int u_{l,m}^*(x)\mathrm{d}\rho(x) = 0$ for all $l \geq 1$, and therefore $\Gamma_i = 0$ for all $i \geq 1$. Therefore (R) holds with $s = +\infty$, and there are no requirements on the eigengaps.

In addition, Lemma 3 of Minh et al. [2006] shows that the supremum-norm of a spherical harmonic is upper bounded by

$$\|u_{l,m}^*\|_\infty \leq \sqrt{\frac{N_l}{|\mathbb{S}^{p-1}|}} = \mathcal{O}\left(i^{\frac{p-2}{2}}\right). \tag{12}$$

The eigenvalue decay rates are obtained from Propositions 2.3 and 2.4 of Scetbon and Harchaoui [2021], and given (12), the condition $a > (p^2 - 4p + 5)/2$ ensures that the conditions for (P) are met.

## B   Proof of Equation (5)

We begin this section with two upper bounds on polynomial and exponential series, which we prove in Section B.1, and which we will use throughout this proof.

**Lemma 2.** *Let $\alpha > 1$, $\beta > 0$ and $0 < \gamma \leq 1$ for fixed constants. Then the following upper bounds hold:*

$$\sum_{i=d+1}^{\infty} i^{-\alpha} = \mathcal{O}\left(d^{-\alpha+1}\right), \qquad \sum_{i=d+1}^{\infty} e^{-\beta i^\gamma} = \mathcal{O}\left(e^{-\beta d^\gamma} d^{1-\gamma}\right).$$

Throughout this proof, $\varepsilon > 0$ will denote some constant which may change from line to line, and even within lines.

To show equation (5), we first note that under (P) with $d = \Omega(n^{1/\alpha})$

$$\sum_{l=d+1}^{n} \lambda_l \lesssim \sum_{i=d+1}^{n} i^{-\alpha} \lesssim d^{-\alpha+1} = \mathcal{O}\left(n^{\frac{-\alpha-1}{\alpha}}\right)$$

where we have used Lemma 2. In addition, under (E) with $d > \log^{1/\gamma}(n^{1/\beta})$, we have that

$$\sum_{l=d+1}^{n} \lambda_l \lesssim \sum_{l=d+1}^{n} e^{-\beta l^\gamma} \lesssim e^{-\beta d^\gamma} d^{1-\gamma} = n^{-(1+\varepsilon)} \log^{(1-\gamma)/\gamma}(n^{1/\beta}) = \mathcal{O}(n^{-1})$$

where we have again used Lemma 2. Now, by the triangle inequality we have that

$$\frac{1}{n} \sum_{l=d+1}^{n} \left| \widehat{\lambda}_l \right| \leq \sum_{l=d+1}^{n} \lambda_l + \sum_{l=d+1}^{n} \left| \frac{\widehat{\lambda}_l}{n} - \lambda_l \right|$$

and therefore we are left to show that

$$\sum_{l=d+1}^{n} \left| \frac{\widehat{\lambda}_l}{n} - \lambda_l \right| = \begin{cases} \mathcal{O}\left(n^{-(\alpha-1)/\alpha} \log(n)\right) & \text{under (P) with } d = \Omega\left(n^{1/\alpha}\right); \\ \mathcal{O}\left(n^{-1}\right) & \text{under (E) with } d > \log^{1/\gamma}(n^{1/\beta}). \end{cases} \tag{13}$$

To bound (13), we employ a fine-grained concentration bound due to Valdivia [2018]. We begin with the polynomial hypothesis. The authors only consider the cases that $\alpha, r$ are natural numbers, since they draw a comparison between between these values and a Sobolev-type notion of regularity. Inspecting their proofs, they treat the cases $r = 0$ and $r \geq 1$ separately, however their proofs follow through in exactly the same way when the $r \geq 1$ case is replaced with $r > 0$, in order to cover all values of $\alpha > 2r + 1$, $r \geq 0$. For the $r > 0$ case, they derive the following result.

**Lemma 3.** *Suppose that the hypothesis (P) holds with $r > 0$. Then, with overwhelming probability*

$$\left| \frac{\widehat{\lambda}_i}{n} - \lambda_i \right| \lesssim B(i, n) \log(n)$$

*where*

$$B(i, n) = \begin{cases} i^{-\alpha + \frac{\alpha}{\alpha-1}(r+\frac{1}{2})} n^{-1/2} & \text{if } 1 \leq i \leq n^{\frac{\alpha-1}{\alpha}\frac{1}{2r+1}}; \\ i^{-\alpha + 1 + \frac{\alpha-1}{\alpha}(r+\frac{1}{2})} n^{-1/2} & \text{if } n^{\frac{\alpha-1}{\alpha}\frac{1}{2r+1}} \leq i \leq n^{\frac{1}{2r}}; \\ i^{-\alpha + r + 1} n^{-1/2} & \text{if } n^{\frac{1}{2r}} \leq i \leq n; \end{cases}$$

Via some rearrangement we can show that

$$B(i, n) = \mathcal{O}\left(i^{-\alpha}\right), \qquad \text{if } 1 \leq i \leq n^{\frac{1}{2r}},$$

and by Lemma 2 we have that

$$\sum_{l=d+1}^{\lfloor n^{1/2r} \rfloor} \left| \frac{\widehat{\lambda}_l}{n} - \lambda_l \right| \lesssim \log(n) \sum_{l=d+1}^{\lfloor n^{1/2r} \rfloor} i^{-\alpha} \lesssim n^{-\frac{\alpha-1}{\alpha}} \log(n). \tag{14}$$

In addition, we have that

$$\begin{aligned}
\sum_{l=\lceil n^{1/2r} \rceil}^{n} \left| \frac{\widehat{\lambda}_l}{n} - \lambda_l \right| &\lesssim n^{-1/2} \log(n) \sum_{l=\lceil n^{1/2r} \rceil}^{n} i^{-\alpha + r + 1} \\
&\lesssim n^{-1/2} \log(n) \cdot \left(n^{\frac{1}{2r}}\right)^{-\alpha + r + 2} \\
&\lesssim n^{-1} \log(n) \\
&\lesssim n^{-\frac{\alpha-1}{\alpha}}
\end{aligned} \tag{15}$$

where we have used that $0 < r < (\alpha - 1)/2$. Combining (14) with (15) establishes (13) under (P) assuming $r > 0$. The case with $r = 0$ follows analogous fashion so we omit the details.

We now turn to the hypothesis (E). The authors only explicitly derive a result for the case that $\gamma = 1$, however following through their proof with Lemma 2 to hand, we obtain the following for the general case that $0 < \gamma \leq 1$.

**Lemma 4.** *Suppose that the hypothesis (E) holds with $s > 0$. Then, with overwhelming probability*

$$\left| \frac{\widehat{\lambda}_i}{n} - \lambda_i \right| \lesssim e^{(-\beta + \delta) i^\gamma} i^{1-\gamma} n^{-1/2} \log(n)$$

*for all $1 \leq i \leq n$.*

Therefore we have that

$$\sum_{i=d+1}^{\infty} \left| \frac{\widehat{\lambda}_i}{n} - \lambda_i \right| \lesssim n^{-1/2} \log(n) \sum_{i=d+1}^{n} e^{(-\beta+s)i} i^{1-\gamma}$$

$$\leq n^{-1/2} \log(n) \sum_{i=d+1}^{n} e^{-(\beta/2+\varepsilon)i^{\gamma}} i^{1-\gamma}$$

$$\lesssim n^{-1/2} \log(n) \sum_{i=d+1}^{n} e^{-(\beta i^{\gamma}/2+\varepsilon)}$$

$$\lesssim n^{-1/2} \log(n) n^{-(1/2+\varepsilon)}$$

$$= \mathcal{O}(n^{-1})$$

where in the second inequality we have used the assumption that $s < \beta/2$ and in the fourth we have used Lemma 2. The case for $s = 0$ follows similarly, so we omit the details. Then Equation (5) is established.

### B.1    Proof of Lemma 2

To bound the polynomial series, we upper bound it by an integral as

$$\sum_{i=d+1}^{\infty} i^{-\alpha} \leq \int_{d}^{\infty} t^{-\alpha} \mathrm{d}t = \frac{d^{-\alpha+1}}{-\alpha+1} = \mathcal{O}\left(d^{-\alpha+1}\right).$$

To bound the exponential series, we again employ an integral approximation, and upper bound it as

$$\sum_{i=d+1}^{\infty} e^{-\beta i^{\gamma}} \leq \int_{d}^{\infty} e^{-\beta t^{\gamma}} \mathrm{d}t.$$

We then apply the substitution $u = \beta t^{\gamma}$ to obtain

$$\int_{d}^{\infty} e^{-\beta t^{\gamma}} \mathrm{d}t = \frac{1}{\gamma} \int_{\beta d^{\gamma}}^{\infty} e^{-u} u^{(1-\gamma)/\gamma} \mathrm{d}u = \frac{1}{\gamma} \Gamma\left(\frac{1}{\gamma}, \beta d^{\gamma}\right)$$

where $\Gamma$ denotes the incomplete Gamma function. We can then use the fact that $\Gamma(s, x) \leq e^{-x} x^{s-1}$ for $s > 0$ to obtain the upper bound

$$\frac{1}{\gamma} \Gamma\left(\frac{1}{\gamma}, \beta d^{\gamma}\right) \leq \frac{1}{\gamma} e^{-\beta d^{\gamma}} (\beta d^{\gamma})^{1/\gamma - 1} = \frac{\beta}{\gamma} e^{\beta d^{\gamma}} d^{1-\gamma},$$

from which we can conclude that

$$\sum_{i=d+1}^{\infty} e^{-\beta i^{\gamma}} = \mathcal{O}\left(e^{\beta d^{\gamma}} d^{1-\gamma}\right),$$

as required.

## C    Proof of Lemma 1

The proof of Lemma 1 generalises the proof of Lemma 43 of Tao and Vu [2011]. We will make use of the following theorem due to Ledoux [2001] which is a corollary of Talagrand's inequality [Talagrand, 1996].

**Theorem 4** (Talagrand's inequality). *Let $y = (y_1, \ldots, y_n)^{\top}$ be a vector of independent random variables, and let $f : \mathbb{R}^n \to \mathbb{R}$ be a convex 1-Lipschitz function. Then, for all $t \geq 0$,*

$$\mathbb{P}\left(|f(y) - M(f)| \geq t\right) \leq 4 \exp\left(-t^2/16\right)$$

*where $M(f)$ denotes the median of $f$.*

It is easy to verify that the map $y \to \|\pi_H(y)\|$ is convex and 1-Lipschitz, and so by Talagrand's inequality we have that

$$\mathbb{P}\left(|\|\pi_H(y)\| - M(\|\pi_H(y)\|)| \geq t\right) \leq 4\exp\left(-t^2/16\right). \tag{16}$$

For $t \geq 8$, we have that

$$4\exp\left(-(t-4)^2/16\right) \leq 4\exp\left(-t^2/32\right),$$

so to conclude the proof, it suffices to show that

$$|M(\|\pi_H(x)\|) - \sigma\sqrt{q}| \leq 4. \tag{17}$$

Let $\mathcal{E}_+$ denote the event that $\|\pi_H(x)\| \geq \sigma\sqrt{q} + 4$ and let $\mathcal{E}_-$ denote the event that $\|\pi_H(x)\| \leq \sigma\sqrt{q} - 4$. By the definition of a median, (17) is established if we can show that $\mathbb{P}(\mathcal{E}_+) < 1/2$ and $\mathbb{P}(\mathcal{E}_-) < 1/2$.

Let $\varepsilon$ be the mean-zero random vector such that $y = \bar{y} + \varepsilon$, and let $P = (p_{ij})_{1 \leq i,j \leq n}$ be the orthogonal projection matrix onto $H$. We have that $\operatorname{tr} P^2 = \operatorname{tr} P = \sum_i p_{ii} = q$ and $|p_{ii}| \leq 1$. Furthermore

$$\|\pi_H(\varepsilon)\|^2 - \sigma^2 q = \sum_{1 \leq i,j \leq n} p_{ij}\varepsilon_i\varepsilon_j - \sigma^2 q = S_1 + S_2.$$

where $S_1 = \sum_{i=1}^n p_{ii}(\varepsilon_i^2 - \sigma^2)$ and $S_2 = \sum_{1 \leq i \neq j \leq n} p_{ij}\varepsilon_i\varepsilon_j$. We now upper bound the expectations of $S_1^2$ and $S_2^2$ which we will use later on for bounding the probabilities of $\mathcal{E}_+$ and $\mathcal{E}_-$ using Markov's inequality. Before we do, note that since $\varepsilon \in [-\bar{y}, 1 - \bar{y}]$ almost surely, Popoviciu's inequality implies that $\sigma^2 \leq 1/4$. Therefore, we also have that $\mathbb{E}(\varepsilon_i^4) \leq \sigma^2$, and so

$$\begin{aligned}
\mathbb{E}\left(S_1^2\right) &= \sum_{i,j=1}^n p_{ii}p_{jj}\mathbb{E}\left\{\left(\varepsilon_i^2 - \sigma^2\right)\left(\varepsilon_j^2 - \sigma^2\right)\right\} = \sum_{i=1}^n p_{ii}^2\mathbb{E}\left\{\left(\varepsilon_i^2 - \sigma^2\right)^2\right\} \\
&\leq \sum_{i=1}^n p_{ii}^2\left\{\mathbb{E}\varepsilon_i^4 - 2\sigma^2\mathbb{E}\left(\varepsilon_i^2\right) + (\sigma^2)^2\right\} \leq \sum_{i=1}^n p_{ii}^2\left\{\sigma^2 - 2(\sigma^2)^2 + (\sigma^2)^2\right\} \leq \sigma^2 q,
\end{aligned} \tag{18}$$

where the second inequality follows from the independence of $\left(\varepsilon_i^2 - \sigma^2\right)$ and $\left(\varepsilon_j^2 - \sigma^2\right)$ for all $i \neq j$. In addition, we have that

$$\mathbb{E}\left(S_2^2\right) = \mathbb{E}\left(\sum_{i \neq j} p_{ij}\varepsilon_i\varepsilon_j\right)^2 = \sum_{i \neq j} p_{ij}^2\mathbb{E}(\varepsilon_i^2)\mathbb{E}(\varepsilon_j^2) \leq (\sigma^2)^2 q \leq \frac{\sigma^2 q}{4}. \tag{19}$$

To upper bound the probability of $\mathcal{E}_+$, we first observe that by assumption

$$\|\pi_H(y)\|^2 = \|\pi_H(\bar{y})\|^2 + \|\pi_H(\varepsilon)\|^2 \leq 4\sigma\sqrt{q} + \|\pi_H(\varepsilon)\|^2$$

and therefore we have that

$$\mathbb{P}(\mathcal{E}_+) = \mathbb{P}\left(\|\pi_H(y)\| \geq \sigma\sqrt{q} + 4\right) \leq \mathbb{P}\left(\|\pi_H(y)\|^2 \geq \sigma^2 q + 8\sigma\sqrt{q}\right) \leq \mathbb{P}\left(\|\pi_H(\varepsilon)\|^2 \geq \sigma^2 q + 4\sigma\sqrt{q}\right).$$

Using the definitions of $S_1$ and $S_2$, it follows that

$$\mathbb{P}(\mathcal{E}_+) \leq \mathbb{P}\left(S_1 + S_2 \geq 4\sigma\sqrt{q}\right) \leq \mathbb{P}\left(S_1 \geq 2\sigma\sqrt{q}\right) + \mathbb{P}\left(S_2 \geq 2\sigma\sqrt{q}\right)$$

By Markov's inequality, we have that

$$\mathbb{P}\left(S_1 \geq 2\sigma\sqrt{q}\right) = \mathbb{P}\left(S_1^2 \geq 4\sigma^2 q\right) \leq \frac{\mathbb{E}\left(S_1^2\right)}{4\sigma^2 q} \leq \frac{1}{4}$$

and similarly that

$$\mathbb{P}\left(S_2 \geq 2\sigma\sqrt{q}\right) = \mathbb{P}\left(S_2^2 \geq 4\sigma^2 q\right) \leq \frac{\mathbb{E}\left(S_2^2\right)}{4\sigma^2 q} \leq \frac{1}{16}.$$

It therefore follows that $\mathbb{P}(\mathcal{E}_+) < 1/2$. We upper bound $\mathbb{P}(\mathcal{E}_-)$ in a similar fashion. Since $\|\pi_H(x)\| \geq \|\pi_H(\varepsilon)\|$, we have that

$$\mathbb{P}\left(\mathcal{E}_-\right) = \mathbb{P}(\|\pi_H(y)\| \leq \sigma\sqrt{q} - 4) \leq \mathbb{P}(\|\pi_H(\varepsilon)\| \leq \sigma\sqrt{q} - 4) = \mathbb{P}(\|\pi_H(\varepsilon)\|^2 \leq \sigma^2 q - 8\sigma\sqrt{q} + 16).$$

Again, recalling the definitions of $S_1$ and $S_2$ we have that

$$\mathbb{P}\left(\mathcal{E}_-\right) \leq \mathbb{P}\left(S_1 + S_2 \leq -8\sigma\sqrt{q} + 16\right) \leq \mathbb{P}(S_1 \leq 8 - 4\sigma\sqrt{q}) + \mathbb{P}(S_2 \leq 8 - 4\sigma\sqrt{q}).$$

As before, applying Markov's inequality we have that

$$\mathbb{P}(S_1 \leq 8 - 4\sigma\sqrt{q}) \leq \mathbb{P}(S_1^2 \geq 64 - 64\sigma\sqrt{q} + 16\sigma^2 q) \leq \mathbb{P}\left(S_1^2 \geq 8\sigma^2 q\right) \leq \frac{\mathbb{E}\left(S_1^2\right)}{8\sigma^2 q} \leq \frac{1}{8}$$

where we have twice used the assumption that $q \geq 64/\sigma^2$. Similarly

$$\mathbb{P}(S_2 \leq 8 - 4\sigma\sqrt{q}) \leq \mathbb{P}\left(S_2^2 \geq 8\sigma^2 q\right) \leq \frac{\mathbb{E}\left(S_2^2\right)}{8\sigma^2 q} \leq \frac{1}{32}.$$

It therefore follows that $\mathbb{P}\left(\mathcal{E}_-\right) < 1/2$, thereby establishing (17) and concluding the proof.

# D   Proof of the claim that $\|\pi_J(\bar{y})\| \leq 2|J|^{1/4}$

In this section, we prove the claim made in the proof of Theorem 1 that

$$\|\pi_J(\bar{y})\| \leq 2|J|^{1/4}, \tag{20}$$

with overwhelming probability, which we require in order to invoke Lemma 1. For notational convenience, we shall assume we are working in an $n$-dimensional space, rather than an $(n-1)$-dimensional space as this is immaterial in our big-$\mathcal{O}$ bounds.

Recall that $\bar{y}$ is a constant vector with entries in $[0, 1]$, and let $\mathbf{1}$ denote the all-ones vector. Let $\xi$ be some value such that $8a < \xi < b/2$, which exists by assumption (R), and let $d'$ denote the smallest index such that

$$\lambda_{d'+1} = \mathcal{O}\left(n^{-1/\xi}\right).$$

This implies that under (P), $d' = \mathcal{O}\left(n^{1/\xi\alpha}\right)$, and under (E), $d' < \log^{1/\gamma}(n^{1/\xi\beta})$. Clearly $d' \leq d$ and so $J \subset \{d'+1, \ldots, n\}$, and therefore

$$\|\pi_J(\bar{y})\| \leq \left\|\pi_{\{d'+1,\ldots,n\}}(\mathbf{1})\right\|.$$

In addition, we observe that

$$\left\|\pi_{\{d'+1,\ldots,n\}}(\mathbf{1})\right\|^2 = n - \left\|\pi_{\{1,\ldots,d'\}}(\mathbf{1})\right\|^2.$$

Since $|J| = \Omega(n)$, it will suffice to show that

$$\frac{1}{n}\left\|\pi_{\{1,\ldots,d'\}}(\mathbf{1})\right\|^2 = 1 - \Omega\left(n^{-1/2-\Omega(1)}\right) \tag{21}$$

with overwhelming probability.

Let $\widehat{U}_{d'}$ and $U_{d'}$ denote the $n \times d'$ matrices with entries

$$\widehat{U}_{d'}(i, j) = \widehat{u}_j(i), \qquad U_{d'}(i, j) = \frac{u_j(x_i)}{n^{1/2}}$$

respectively. Then we have

$$\frac{1}{n}\left\|\pi_{\{1,\ldots,d'\}}(\mathbf{1})\right\|^2 = \frac{1}{n}\left\|\widehat{U}_{d'}\widehat{U}_{d'}^\top\mathbf{1}\right\|^2 = \frac{1}{n}\left\|\widehat{U}_{d'}^\top\mathbf{1}\right\|^2 = \frac{1}{n}\left\|(\widehat{U}_{d'}W)^\top\mathbf{1}\right\|^2,$$

where $W$ is an orthogonal matrix which we will define later on. By the triangle inequality, we have that

$$\frac{1}{n}\left\|(\widehat{U}_{d'}W)^\top\mathbf{1}\right\|^2 \geq \frac{1}{n}\left\|U_{d'}^\top\mathbf{1}\right\|^2 - \frac{1}{n}\left\|\widehat{U}_{d'}W - U_{d'}^\top\right\|^2\|\mathbf{1}\|^2 = \frac{1}{n}\left\|U_{d'}^\top\mathbf{1}\right\|^2 - \left\|\widehat{U}_{d'}W - U_{d'}^\top\right\|^2$$

and therefore to show (21), we need to show that

$$n^{-1}\left\|U_{d'}^\top\mathbf{1}\right\|^2 = 1 - \mathcal{O}\left(n^{-1/2-\Omega(1)}\right) \tag{22}$$

and

$$\left\|\widehat{U}_{d'}W - U_{d'}^\top\right\| = O\left(n^{-1/4-\Omega(1)}\right) \tag{23}$$

with overwhelming probability.

## D.1 Bounding (22)

To bound (22), we begin by using the the inequality $(c_1 + c_2)^2 \leq 2(c_1^2 + c_2^2)$ to write

$$n^{-1} \left\| U_{d'}^\top \mathbf{1} \right\|^2 = n^{-1} \sum_{l=1}^{d'} \left( \sum_{i=1}^{n} \frac{u_l(x_i)}{n^{1/2}} \right)^2 = \sum_{l=1}^{d'} \left( \sum_{i=1}^{n} \frac{u_l(x_i)}{n} \right)^2$$

$$\geq \sum_{l=1}^{d'} \left( \int u_l(x)\mathrm{d}\rho(x) \right)^2 - 2 \sum_{l=1}^{d'} \left( \sum_{i=1}^{n} \frac{u_l(x_i)}{n} - \int u_l(x)\mathrm{d}\rho(x) \right)^2.$$

To bound the first term, we observe that since $\{u_i\}_{i=1}^{\infty}$ forms an orthonormal basis for $L_\rho^2(\mathcal{X})$, we have that

$$\sum_{l=1}^{\infty} \left( \int u_l(x)\mathrm{d}\rho(x) \right)^2 = 1$$

and therefore

$$\sum_{l=1}^{d'} \left( \int u_l(x)\mathrm{d}\rho(x) \right)^2 = 1 - \sum_{l=d'+1}^{\infty} \left( \int u_l(x)\mathrm{d}\rho(x) \right)^2 =: 1 - \Gamma_{d'+1}$$

By assumption,

$$\Gamma_{d'+1} = \mathcal{O}\left( \lambda_{d'+1}^b \right) = \mathcal{O}\left( n^{-b/\xi} \right) = \mathcal{O}\left( n^{-1/2 - \Omega(1)} \right)$$

where we have used the assumption that $b > \xi/2$, and therefore

$$\sum_{l=1}^{d'} \left( \int u_l(x)\mathrm{d}\rho(x) \right)^2 = 1 - \mathcal{O}\left( n^{-b/\xi} \right) = 1 - \mathcal{O}\left( n^{-1/2 - \Omega(1)} \right),$$

as required.

Now to bound the second term, we use Hoeffding's inequality to obtain that

$$\left| \sum_{i=1}^{n} \frac{u_l(x_i)}{n} - \int u_l(x)\mathrm{d}\rho(x) \right| = \mathcal{O}\left( \|u_l\|_\infty \frac{\log^{1/2}(n)}{n^{1/2}} \right).$$

Under (P), we have that for all $l \leq d'$,

$$\|u_l\|_\infty \lesssim d'^r \lesssim n^{r/\xi\alpha} = \mathcal{O}\left( n^{1/2\xi} \right) = \mathcal{O}(n^{1/16 - \Omega(1)})$$

where we have used that $r < (\alpha - 1)/2 \leq \alpha/2$, and that $\xi > 8$, and under (E)

$$\|u_l\|_\infty \lesssim e^{sd'^\gamma} \lesssim e^{s \log(n)/\xi\beta} = \mathcal{O}\left( n^{1/2\xi} \right) = \mathcal{O}(n^{1/16 - \Omega(1)})$$

where we have used that $s < \beta/2$. Therefore, since $d' = \mathcal{O}\left( n^{1/8} \right)$, we have that

$$\sum_{l=1}^{d'} \left( \sum_{i=1}^{n} \frac{u_l(x_i)}{n} - \int u_l(x)\mathrm{d}\rho(x) \right)^2 \lesssim d' \left( n^{1/16 - 1/2} \right)^2 = \mathcal{O}\left( n^{-3/4} \right)$$

which is $\mathcal{O}\left( n^{-1/2 - \Omega(1)} \right)$ as required.

## D.2 Bounding (23)

To bound (23), we begin by defining the $d' \times d'$ diagonal matrices $\widehat{\Lambda}_{d'}$ and $\Lambda_{d'}$ with diagonal entries

$$\widehat{\Lambda}_{d'}(i, i) := \frac{\widehat{\lambda}_i}{n}, \qquad \Lambda_{d'}(i, i) := \lambda_i,$$

respectively, and the $n \times d'$ matrices $\widehat{\Phi}_{d'}$ and $\Phi_{d'}$ with entries

$$\widehat{\Phi}_{d'}(i, j) = \widehat{\lambda}_j^{1/2} \widehat{u}_j(i), \qquad \Phi_{d'}(i, j) = \lambda_j^{1/2} u_j(i),$$

respectively. Then in matrix notation we have that

$$\widehat{\Phi}_{d'} = n^{1/2}\widehat{U}_{d'}\widehat{\Lambda}_{d'}^{1/2}, \qquad \Phi_{d'} = n^{1/2}U_{d'}\Lambda_{d'}^{1/2}$$

Now, we decompose $\widehat{U}_{d'}W_{d'} - U_{d'}$ as

$$\widehat{U}_{d'}W_{d'} - n^{-1/2}U_{d'} = \left\{ n^{-1/2}\left(\widehat{\Phi}_{d'}W_{d'} - \Phi_{d'}\right) + \widehat{U}_{d'}\left(W_{d'}\Lambda_{d'}^{1/2} - \widehat{\Lambda}_{d'}^{1/2}W_{d'}\right)\right\}\Lambda^{-1/2}$$

and so we have that

$$\left\|\widehat{U}_{d'}W_{d'} - U_{d'}\right\| \le \lambda_{d'}^{-1/2}\left\{n^{-1/2}\left\|\widehat{\Phi}_{d'}W_{d'} - \Phi_{d'}\right\|_{\mathrm{F}} + \left\|W_{d'}\Lambda_{d'}^{1/2} - \widehat{\Lambda}_{d'}^{1/2}W_{d'}\right\|\right\}.$$

By the construction of $d'$ we have that $\lambda_{d'} = \Omega(n^{-1/\xi}) = \Omega(n^{-1/8})$ and therefore $\lambda_{d'}^{-1/2} = \mathcal{O}\left(n^{1/16}\right)$. Therefore to show (23), it will suffice to show that

$$\left\|\widehat{\Phi}_{d'}W_{d'} - \Phi_{d'}\right\|_{\mathrm{F}} = \mathcal{O}\left(n^{\frac{3}{16}-\Omega(1)}\right) \tag{24}$$

and

$$\left\|W_{d'}\Lambda_{d'}^{1/2} - \widehat{\Lambda}_{d'}^{1/2}W_{d'}\right\| = \mathcal{O}\left(n^{-\frac{5}{16}-\Omega(1)}\right). \tag{25}$$

To obtain the bound (24), we make use of the following result which is Equation 3.8 of Tang et al. [2013].

**Lemma 5.** *The exists an orthogonal matrix $W_d$ (constructed as in (29)) such that*

$$\left\|\widehat{\Phi}_d W_d - \Phi_d\right\|_{\mathrm{F}} = \mathcal{O}\left(\frac{\log^{1/2}(n)}{\lambda_d - \lambda_{d+1}}\right)$$

*with overwhelming probability.*

Now, let

$$d^\star := \arg\max_{i \ge d'}\left\{\lambda_i - \lambda_{i+1}\right\}, \tag{26}$$

then by the assumption (R), we have that

$$\lambda_{d^\star} - \lambda_{d^\star+1} = \Omega(\lambda_{d'}^a) = \Omega(n^{-a/\xi}) = \Omega\left(n^{-1/8+\Omega(1)}\right). \tag{27}$$

Using Lemma 5, we obtain that

$$\left\|\widehat{\Phi}_{d'}W_{d'} - \Phi_{d'}\right\|_{\mathrm{F}} \le \left\|\widehat{\Phi}_{d^\star}W_{d^\star} - \Phi_{d^\star}\right\|_{\mathrm{F}} = \mathcal{O}\left(\frac{\log^{1/2}(n)}{\lambda_{d^\star} - \lambda_{d^\star+1}}\right) = \mathcal{O}\left(n^{1/8-\Omega(1)}\right),$$

which establishes (24). Obtaining the bound (25) requires some new concepts, so we dedicate it its own section.

### D.3 Bounding (25)

We now turn our attention to the bound (25). We begin by defining $\mathcal{H}$, the reproducing kernel Hilbert space associated with the kernel $k$, and define the operators $\mathcal{K}_{\mathcal{H}}, \widehat{\mathcal{K}}_{\mathcal{H}} : \mathcal{H} \to \mathcal{H}$ given by

$$\mathcal{K}_{\mathcal{H}}f = \int \langle f, k_x\rangle_{\mathcal{H}} k_x \mathrm{d}\rho(x),$$
$$\widehat{\mathcal{K}}_{\mathcal{H}} = \frac{1}{n}\sum_{i=1}^{n}\langle f, k_{x_i}\rangle_{\mathcal{H}} k_{x_i} \tag{28}$$

where $k_x(\cdot) = k(\cdot, x)$ and $\langle \cdot, \cdot\rangle_{\mathcal{H}}$ is the inner product in $\mathcal{H}$. These operators are known as the "extension operators" of $\mathcal{K}$ and $\frac{1}{n}K$, respectively, and it may be shown that each has the same eigenvalues as its corresponding operator, possibly up to zeros (see e.g. Propositions 8 and 9 of Rosasco et al. [2010]). We will make use of the following concentration inequality for $\mathcal{K}_{\mathcal{H}} - \widehat{\mathcal{K}}_{\mathcal{H}}$ which is due to De Vito et al. [2005] and appears as Theorem 7 of Rosasco et al. [2010].

**Lemma 6** (Theorem 7 of Rosasco et al. [2010]). *The operators $\mathcal{K}_{\mathcal{H}}$ and $\widehat{\mathcal{K}}_{\mathcal{H}}$ are Hilbert-Schmidt, and*

$$\left\|\widehat{\mathcal{K}}_{\mathcal{H}} - \mathcal{K}_{\mathcal{H}}\right\|_{HS} = \mathcal{O}\left(\frac{\log^{1/2}(n)}{n^{1/2}}\right)$$

*with overwhelming probability.*

Let $\{u_{\mathcal{H},i}\}_{i \leq d}$ denote the eigenfunctions of $\mathcal{K}_{\mathcal{H}}$ corresponding to the eigenvalues $\{\lambda_i\}_{i \leq d}$, and let $\{\widehat{u}_{\mathcal{H},i}\}_{i \leq d}$ denote the eigenfunctions of $\widehat{\mathcal{K}}$ corresponding to the eigenvalues $\{\widehat{\lambda}_i/n\}_{i \leq d}$. We define the (infinite-dimensional) "matrices" $U_{\mathcal{H},d}$ and $\widehat{U}_{\mathcal{H},d}$ whose columns contain the eigenfunctions $\{u_{\mathcal{H},i}\}_{i \leq d}$ and $\{\widehat{u}_{\mathcal{H},i}\}_{i \leq d}$, respectively. These "matrices" are well-defined with matrix multiplication is defined as usual with inner products taken in $\mathcal{H}$. We will refer to $U_{\mathcal{H},d}, \widehat{U}_{\mathcal{H},d}$ and the subspaces spanned by their columns interchangably.

Let $H_d = U_{\mathcal{H},d}^\top \widehat{U}_{\mathcal{H},d}$ with entries $H_d(i,j) = \langle u_{\mathcal{H},i}, \widehat{u}_{\mathcal{H},j}\rangle_{\mathcal{H}}$ and denote its singular values by $\xi_1, \ldots, \xi_d$. Then the principal angles between the subspaces $U_{\mathcal{H},d}$ and $\widehat{U}_{\mathcal{H},d}$, which we will denote by $\theta_1, \ldots, \theta_d$, are define as via $\xi_i = \cos(\theta_i)$. Define the matrix

$$\sin\Theta\left(\widehat{U}_{\mathcal{H},d}, U_{\mathcal{H},d}\right) = \text{diag}\left(\sin(\theta_1), \ldots, \sin(\theta_d)\right).$$

Let $d^\star$ be as in (26) so that $\lambda_{d^\star} - \lambda_{d^\star+1} = \Omega\left(n^{-1/8+\Omega(1)}\right)$. Since $d^\star \geq d'$, by the Davis-Kahan theorem, we have that

$$\left\|\sin\Theta\left(\widehat{U}_{\mathcal{H},d'}, U_{\mathcal{H},d'}\right)\right\| \leq \left\|\sin\Theta\left(\widehat{U}_{\mathcal{H},d^\star}, U_{\mathcal{H},d^\star}\right)\right\| \leq \frac{\sqrt{2}\left\|\widehat{\mathcal{K}}_{\mathcal{H}} - \mathcal{K}_{\mathcal{H}}\right\|}{\lambda_{d^\star} - \lambda_{d^\star+1}}$$

$$\leq \frac{\sqrt{2}\left\|\widehat{\mathcal{K}}_{\mathcal{H}} - \mathcal{K}_{\mathcal{H}}\right\|_{HS}}{\lambda_{d^\star} - \lambda_{d^\star+1}} = \mathcal{O}\left(n^{-3/8-\Omega(1)}\right).$$

where the second inequality comes from the relationship $\|\cdot\| \leq \|\cdot\|_{HS}$, and the final inequality follows from Lemma 6 and (27).

We now come to constructing the matrix $W_d$. We denote the singular value decomposition of $H$ as $H = W_{d,1} \Xi W_{d,2}^\top$ and define the matrix $W_d$ by

$$W = W_{d,1} W_{d,2}^\top, \tag{29}$$

which is known as the "matrix sign" of $H$. Let $\widehat{\mathcal{P}}_d$ and $\mathcal{P}_d$ to be the projections onto the subspaces $\widehat{U}_{\mathcal{H},d}$ and $U_{\mathcal{H},d}$, respectively. Then we have the following decomposition.

$$\begin{aligned}
W_{d'} \Lambda_{d'}^{1/2} - \widehat{\Lambda}_{d'}^{1/2} W_{d'} &= (W_{d'} - H_{d'})\widehat{\Lambda}_{d'} + \Lambda_{d'}(H_{d'} - W_{d'}) \\
&\quad + U_{\mathcal{H},d'}^\top(\widehat{\mathcal{K}}_{\mathcal{H}} - \mathcal{K}_{\mathcal{H}})(\widehat{\mathcal{P}}_{d'} - \mathcal{P}_{d'})\widehat{U}_{\mathcal{H},d'} \\
&\quad + U_{\mathcal{H},d'}^\top(\widehat{\mathcal{K}}_{\mathcal{H}} - \mathcal{K}_{\mathcal{H}})U_{\mathcal{H},d'} H_{d'}
\end{aligned} \tag{30}$$

We first observe that $\|H_{d'}\| \leq \|\widehat{U}_{\mathcal{H},d'}\|\|U_{\mathcal{H},d'}\| = 1$, and by (5), $\left\|\widehat{\Lambda}_{d'}\right\|, \|\Lambda_{d'}\| = \mathcal{O}(1)$. In addition, following the same steps as in the proof Lemma 6.7 of Cape et al. [2019] (who prove similar (in)equalities for finite-dimensional matrices) we have that

$$\left\|\widehat{\mathcal{P}}_{d'} - \mathcal{P}_{d'}\right\| = \left\|\sin\Theta\left(\widehat{U}_{\mathcal{H},d'}, U_{\mathcal{H},d'}\right)\right\| = \mathcal{O}\left(n^{-3/8-\Omega(1)}\right)$$

$$\|H_{d'} - W_{d'}\| \leq \left\|\sin\Theta\left(\widehat{U}_{\mathcal{H},d'}, U_{\mathcal{H},d'}\right)\right\|^2 = \mathcal{O}\left(n^{-3/4-\Omega(1)}\right).$$

We now turn to bounding $\|U_{\mathcal{H},d}^\top(\widehat{\mathcal{K}}_{\mathcal{H}} - \mathcal{K}_{\mathcal{H}})U_{\mathcal{H},d}\|$. To condense notation, let $Q = U_{\mathcal{H},d}^\top(\widehat{\mathcal{K}}_{\mathcal{H}} - \mathcal{K}_{\mathcal{H}})U_{\mathcal{H},d}$. We will bound $\|Q\|$ using a classical $\varepsilon$-net argument. Let $\mathbb{S}_{\varepsilon}^{d-1}$ be an $\varepsilon$-net of the $(d-1)$-dimensional unit sphere $\mathbb{S}^{d-1} := \{v : \|v\| = 1\}$, that is, a subset of $\mathbb{S}^{d-1}$ such that for any $v \in \mathbb{S}^{d-1}$,

there exists some $w_v \in \mathbb{S}_\varepsilon^{d-1}$ such that $\|v - w_v\| < \varepsilon$. Then, we have that

$$
\begin{aligned}
\|Q\| &= \max_{v: \|v\| \leq 1} \left| v^\top Q v \right| \\
&= \max_{v: \|v\| \leq 1} \left| (v - w_v + w_v)^\top Q (v - w_v + w_v) \right| \\
&\leq (\varepsilon^2 + 2\varepsilon) \|Q\| + \max_{w \in \mathbb{S}_\varepsilon^{d-1}} \left| w^\top Q w \right|.
\end{aligned}
$$

With $\varepsilon = 1/3$, we have

$$
\|Q\| \leq \frac{9}{2} \max_{w \in \mathbb{S}_\varepsilon^{d-1}} \left| w^\top Q w \right|.
$$

Using the definitions (28), its $(l, m)$th entry can be calculated as

$$
Q(l, m) = \frac{1}{n^2} \sum_{i,j=1}^n k(x_i, x_j) u_l(x_i) u_m(x_j) - \iint k(x, y) u_l(x) u_m(y) \mathrm{d}\rho(x) \mathrm{d}\rho(y), \qquad (31)
$$

and so for a given $w \in \mathbb{S}_\varepsilon^{d-1}$, we have that

$$
\begin{aligned}
\left| w^\top Q w \right| &= \left| \sum_{l,m=1}^d Q(l, m) w(l) w(m) \right| \\
&= \left| \sum_{l,m=1}^d w(l) w(m) \left( \frac{1}{n^2} \sum_{i,j=1}^n k(x_i, x_j) u_l(x_i) u_m(x_j) - \iint k(x, y) u_l(x) u_m(y) \mathrm{d}\rho(x) \mathrm{d}\rho(y) \right) \right| \\
&\leq \max_{1 \leq l \leq d} \|u_l\|_\infty \left| \sum_{i=1}^n \left\{ \sum_{l,m=1}^d w(l) w(m) \left( \frac{u_m(x_i)}{n} - \int u_m(x) \mathrm{d}\rho(x) \right) \right\} \right|
\end{aligned}
$$

where we have used that $k(x, y) \leq 1$ for all $x, y \in \mathcal{X}$. This is a sum of independent random variables, so by Hoeffding's inequality, we that that

$$
\mathbb{P}\left( \left| w^\top Q w \right| \geq t \right) \leq 2 \exp\left( -\frac{2nt^2}{\max_{1 \leq l \leq d} \|u_l\|_\infty^4} \right)
$$

where we have used that $\sum_{l,m=1}^d w(l) w(m) = 1$. The set $\mathbb{S}_{1/3}^{d-1}$ can be selected so that its cardinality is no greater than $18^d$ (see, for example, Pollard [1990]), so using a union bound we have that

$$
\begin{aligned}
\mathbb{P}\left( \|Q\| \geq t \right) &\leq \mathbb{P}\left( \max_{w \in \mathbb{S}_{1/3}^{d-1}} \left| w^\top Q w \right| \geq \frac{2}{9} t \right) \\
&\leq \sum w \in \mathbb{S}_{1/3}^{d-1} \mathbb{P}\left( \left| w^\top Q w \right| \geq \frac{2}{9} t \right) \\
&\leq 2 \cdot 18^{d'} \exp\left( -\frac{8nt^2}{81 \cdot \max_{1 \leq l \leq d'} \|u_l\|_\infty^4} \right) \\
&\leq 2 \exp\left( d' \log(18) - \frac{8nt^2}{81 \cdot \max_{1 \leq l \leq d'} \|u_l\|_\infty^4} \right) \\
&\leq 2 \exp\left( C \left( n^{1/8 - \Omega(1)} - n^{3/4} t^2 \right) \right)
\end{aligned}
$$

where we have used that $d' = \mathcal{O}(n^{1/8 - \Omega(1)})$ and $\max_{1 \leq l \leq d'} \|u_l\|_\infty = \mathcal{O}(n^{1/16})$. Choosing $t = 2n^{-5/16 - \Omega(1)}$ we have that

$$
\mathbb{P}\left( \|Q\| \geq 2n^{-5/16 - \Omega(1)} \right) \leq 2 \exp\left( -n^{1/8} \right) \leq n^{-c}
$$

for any $c > 0$ for large enough $n$. Therefore

$$
\|Q\| = \mathcal{O}\left( n^{-5/16 - \Omega(1)} \right)
$$

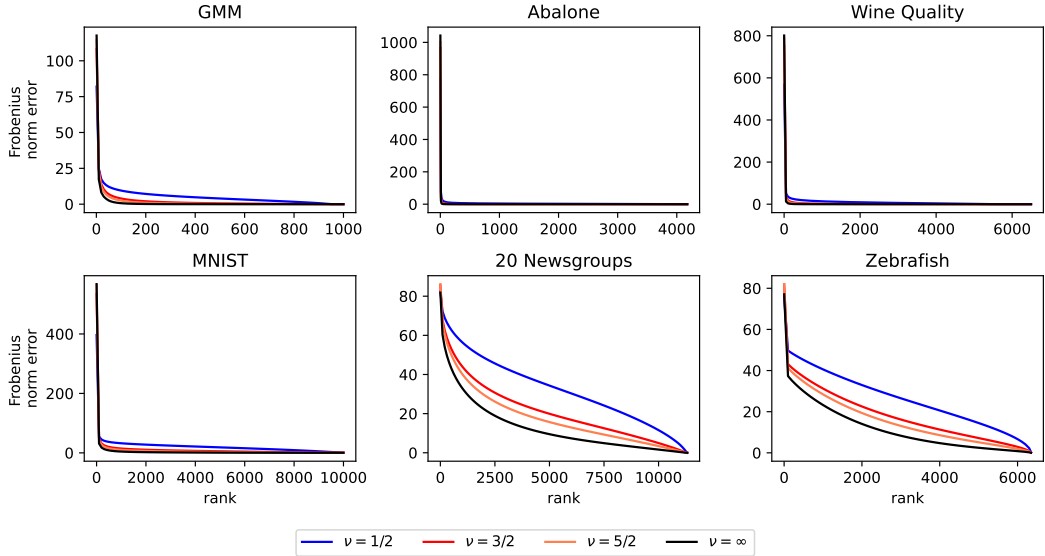

Figure 2: The Frobenius-norm error against rank for low-rank approximations of kernel matrices constructed from a collection of datasets. The kernel matrices are constructed using Matérn kernels with a range of smoothness parameters, each of which is represented by a line in each plot. Details of the experiment are provided in Section 5.

with overwhelming probability.

Combining the above bounds with (30) we obtain that

$$\left\| W_{d'} \Lambda_{d'}^{1/2} - \widehat{\Lambda}_{d'}^{1/2} W_{d'} \right\| = \mathcal{O}\left( n^{-5/16 - \Omega(1)} \right)$$

which establishes (25) and therefore completes the proof.

# E    Additional details about the experiments

In this section, we provide some additional details and plots relating to the experiments in Section 5.

### E.1    Details about the datasets

*GMM* is a synthetic dataset of 1,000 simulated data points from a 10-component Gaussian mixture model with unit isotropic covariances and means of size ten on the axes. *Abalone* [Nash et al., 1995] and *Wine Quality* [Cortez et al., 2009] are popular benchmark datasets which we standarised in the usual way by centering and rescaling each feature to have unit variance. We drop the binary *Sex* feature from the *Abalone* dataset to retain only the continuous features. *MNIST* [Deng, 2012] is a dataset of handwritten digits, represented as 28x28 gray-scale pixels which we concatenate into 784 dimensional vectors. These four datasets might be described as "low-dimensional", and are thus representative of the theory we present in Section 3.

In addition, we consider two "high-dimensional" datasets. *20 Newsgroups* [Lang, 1995] is a popular natural language dataset of messages collected from twenty different "newnews" newsgroups. We remove stop-word and words which appear in fewer than 5 documents or more than 80% of them, and convert each document into a vector using term frequency-inverse document frequency (tf-idf) features for each word. *Zebrafish* [Wagner et al., 2018] is a dataset of single-cell gene expression in a zebrafish embryo taken during their first day of development. We subsample 10% of the cells, and process the data following the steps in Wagner et al. [2018].

## E.2 Frobenius norm errors

Figure 2 shows the Frobenius norm error of the low-rank approximations. For the low-dimensional datasets *GMM*, *Abalone*, *Wine Quality* and *MNIST*, the Frobenius norm error decays very quickly. However for the high-dimensional datasets *20 Newsgroups* and *Zebrafish*, the Frobenius norm error decays much more slowly. As pointed out in the main text, the Frobenius norm error of the *20 Newsgroups* dataset does not exhibit the sharp drop between $d = 2500$ and $d = 3000$ that the maximum entrywise error exhibits.

## E.3 Implementation details

The experiments were performed on the HPC cluster at Imperial College London with 8 cores and 16GB of RAM. The *GMM* experiment took less than 1 minute; the *Abalone*, *Wine Quality*, *MNIST* and *Zebrafish* experiments took less than 8 hours; and the *20 Newsgroups* experiment took less than 24 hours.

