# OpenReview forum: "Entrywise error bounds for low-rank approximations of kernel matrices"
_NeurIPS.cc/2024/Conference — NeurIPS 2024 poster_

### Official Review · Reviewer_2JjA · 2024-07-04

**Soundness:** 3
**Presentation:** 3
**Contribution:** 3
**Rating:** 6
**Confidence:** 3

**Summary:**

This paper is first to establish entrywise guarantees for low rank approximation of kernel matrices when kernel eigenvalues satisfy either polynomial or exponential decay. More specifically, in the $\alpha$-polynomial decay setting, entrywise error scales as $O(n^{-\\frac{\alpha-1}{\\alpha}} \\log n)$ for rank $d = \Omega(n^{1/\\alpha})$, while for $(\\beta,\\gamma)$-exponential decay error scales like $O(1/n)$ for $d > \\log^{1/\\gamma}(n^{1/\\beta})$. In order to establish such results, authors prove that eigenvectors corresponding to small eigenvalues are completely incoherent/delocalized i.e. have bounded entries of size $O(1/\\sqrt{n})$. Technical novelty stems from the fact that entries of the kernel matrix are dependent and have non-zero mean.

**Strengths:**

1) This is a first result showing entrywise error guarantees for low rank approximation of kernel matrices.

2) Proof sketches of two main theorems are clear and easy to follow.

3) Strongest technical contribution of this paper is proof given in Appendix D that, simply speaking, shows that the norm of projection of vector 1 on the subspace spanned by $n-d'$ eigenvectors with smallest eigenvalues is vanishing sufficiently fast.

4) Experiments are complementing theoretical results well.

**Weaknesses:**

1) Although authors claim that Lemma 1 is a novel concentration result, it seems to be only a slight generalization of Lemma 68 in Tao and Vu [2011], and is proved essentially using the same argument as that in the proof of Lemma 68.

2) Although I appreciate proof sketches of Theorems 1 and 2 in the main text, I believe it would be more useful to add more information about the proof deferred to Appendix D since this is the most novel and interesting part of the proof.

3) It is not clear whether assumption (R) is necessary and how general it is apart from the two special cases given in Section 3.1.

**Questions:**

1) Could you elaborate more on tightness of your results? How do they compare with already established results for Frobenius and spectral norm? Are there any known lower bounds for entrywise estimation?

2) Although assumptions (E) and (P) seem to be very natural, I am not sure about assumption (R). Do results hold for any $a$ and $b$ such that $1\\leq a < b/16$? Since the final error bound does not depend on $a$ and $b$, do you think this assumption can be relaxed?

3) Although I think that double descent observation is interesting on its own, the evidence for it is vague. Is this behavior observed for a range of percentile values or does it happen only around 99.95 percentile? Also from figures in the paper seem like it appears only for not very smooth kernel functions. It would be beneficial to have more convincing evidence whether this phenomenon occurs because of your choice of 1) kernels, 2) percentiles, 3) entrywise errors or something else.


Typos and other comments

(106) maximum entrywise "error" (missing)

(186) should be $ \\hat{u}_i(1)$, instead of $ \\hat{u}_l(1) $

(647) later on

I would prefer if you do not use $(a,b)$ both for constants in assumption (R) and for vectors in the proof of Theorem 2.

In introduction you cite [Lei, 2019] for establishing entrywise error bounds for reinforcement learning - but I could not find any references to RL in that paper. Is this a typo? For example, I believe the following papers are more suitable for that particular reference:

- Pananjady, Ashwin, and Martin J. Wainwright. "Instance-dependent ℓ∞-bounds for policy evaluation in tabular reinforcement learning." IEEE Transactions on Information Theory 67.1 (2020): 566-585.
- Shah, Devavrat, et al. "Sample efficient reinforcement learning via low-rank matrix estimation." Advances in Neural Information Processing Systems 33 (2020): 12092-12103.
- Stojanovic, Stefan, Yassir Jedra, and Alexandre Proutiere. "Spectral entry-wise matrix estimation for low-rank reinforcement learning." Advances in Neural Information Processing Systems 36 (2023): 77056-77070.

**Limitations:**

The authors have addressed limitations adequately.

---

> ### Author Rebuttal · Authors · 2024-08-05
>
> I thank the reviewer for their time and effort reviewing my paper.
>
> The reviewer argues that Lemma 1 is only a slight generalisation of Lemma 68 in Tao and Vu (2011), although understanding the conditions which one must place on the mean vector when it is non-zero for the lemma to work required some technical innovation, and could not simply be guessed without working through the mathematical details. I provide more details in the global response.
>
> I was happy to read that the reviewer found the proofs in Section D to represent a strong technical contribution and I agree that it deserves more space in the main paper. The main reason that it did not receive it was simply space limitations. If space allows, I will include a sketch of Section D in the main paper in the revision.
>
> In response to the concern about assumption (R), I provide additional details and discussion in the global response.
>
> I included the double descent observation as a curiosity for readers and other researchers, in the hope it may encourage additional investigation into the phenomenon. The phenomenon does occur at lower quantiles albeit less extremely (even at the median in some cases), and I can include some additional plots in the appendix, however I don't plan to investigate this further in this paper.
>
> I thank the reviewer for pointing out some typos which I will correct, and I agree with the comment about using the constants $a$ and $b$ in multiple settings. I will use different letters for the vectors in the proof of Theorem 2. I also thank the reviewer for noticing the incorrect citation in the introduction - it is a typo! I had meant to cite the mentioned Stojanovic et al. paper.
>
> Coming to Question 1, I am not aware of any lower bounds for entrywise estimation in this setting, although this would be an interesting direction for future research. The question about how the bounds compare to naive spectral and Frobenius norm bounds is an interesting question. Indeed, one can naively upper bound the entrywise norm $\|\hat K - K\|_\max$ by the spectral norm $\|\hat K - K\|_2 = \hat \lambda_{d+1}$ (or worse, the Frobenius norm). There are two reasons that this approach does not lead to satisfactory bounds. First if one has a bound of the form
>
> $$
> \hat \lambda_{d+1} = n \lambda_{d+1} + \textsf{approximation error}
> $$
>
> where the approximation error is of the order $O(n\lambda_{d+1})$, then to obtain the same rates as Theorem 1, one would require that $d \geq \log^{1/\gamma}(n^{2/\beta})$ under (E), and $d = \Omega(n^{2/\alpha})$ under (P), which are much larger than the $d$ required in Theorem 1. The second stumbling block is that when $d$ is this large, using the best available concentration inequalities (i.e. Theorem 2 of Valdivia (2018)), the approximation error will in fact dominate $n\lambda_{d+1}$ and in this case it may not even be possible to obtain a shrinking error rate for any $d$​!
>
> *References:*
> - Terence Tao and Van Vu. Random matrices: Universality of local eigenvalue statistics. Acta Mathematica, 206(1):127 – 204, 2011.
> - Ernesto Araya Valdivia. Relative concentration bounds for the spectrum of kernel matrices. arXiv preprint arXiv:1812.02108, 2018.

---

> > ### Comment · Reviewer_2JjA · 2024-08-12
> >
> > Thank you for your reply. I will maintain my score.

---

### Official Review · Reviewer_3p4Z · 2024-07-08

**Soundness:** 3
**Presentation:** 3
**Contribution:** 2
**Rating:** 5
**Confidence:** 3

**Summary:**

The paper focuses on deriving entrywise error bounds for low-rank approximations of kernel matrices using truncated eigen-decomposition. It addresses the statistical behavior of individual entries in such approximations under assumptions of polynomial eigenvalue decay or exponential decay. The authors also provide empirical studies on synthetic and real-world datasets.

**Strengths:**

1. The paper is clear and well written. The proof seems to be solid.
2. The entrywise error bound is new to the community.
3. The assumptions on polynomial/exponential eigenvalue decay seem general and cover lots of common kernels.
4. Some statements about random matrix theory and concentration inequalities are provided (e.g., Lemma 1), which could be independently useful to the community.

**Weaknesses:**

1. The assumptions on the eigenfunctions corresponding to the assumptions of eigenvalue decay are hard to verify for general kernels, especially the part on the rate of decay ($\alpha >2r+1,\beta> 2s$). Moreover, I wonder if these inequalities are required to guanrantee the uniform convergence of the kernel (I note that $k(x,y)=\sum_{i=1}^{\infty}\lambda_i u_i(x)u_i(y)$ converges uniformly under these assumptions).  But in the proof I see these assumptions are used in a way like $\beta-s\ge \beta/2$ (e.g., Line 590). Thus, I am not sure if these asssumptions are necessary for derivation.
2. Assumption (R) seems not natural (why is $1\le a < b/16$ is needed?) and also I do not know how to verify this. Could you provide some examples with $\Gamma_i \neq 0$ under Assumption (R)?
3. The contributions are undetermined. The proof of the main theorem seems to heavily rely on past random matrix theory works (Tao and Vu [2011], Erdős et al. [2009 a,b]). With assumptions (E)/(P) and (R) and the previous works, the proof is straightfoward. And I am not sure about the importance of entrywise error bound.

Minor typos:
1. Line 578/588 hypotheses-> hypothesis
2. Line 539/581 miss a period

**Questions:**

1. (Line 82) What do you mean by "infinite sample limit of $\frac{1}{n}K$"?
2. Could you provide more general examples that completely follow the assumption (E)/(P)?
3. Is this error bound optimal? Are there any lower bound results?
4. Is it possible (or are there any hardness results) to compute or approximate $\text{argmin}_{K':\text{rank}(K')=d}$ $ \||K-K'\||$ w.r.t. the sup norm?
5. Regarding the importance of entrywise error bound, could you provide more concrete examples?

**Limitations:**

There is no negative societal impact of their work.

---

> ### Author Rebuttal · Authors · 2024-08-05
>
> To begin, I would like to thank the reviewer for taking the time to review my paper. They considered the writing and proofs to be clear and accurate, and the theoretical result to be new to the community.
>
> The reviewer mentions that the assumptions on the eigenvalue decay and the eigenfunction growth (assumptions (P) and (E)) are hard to verify for general kernels. Indeed, this is a general shortcoming of many theoretical works on kernel methods which I discuss in the limitations section of the paper, but I will provide some additional context here. As noted by the reviewer, each of these assumptions imply the convergence of the eigendecomposition. Firstly, given the eigenvalue decay condition (either polynomial of exponential), the eigenfunction growth condition is the weakest possible which ensures the convergence of the eigendecomposition. We need explicit decay/growth conditions on the eigenvalues and eigenfunctions for two reasons: firstly, our results rely on a state-of-the-art eigenvalue concentration inequality due to Valdivia (2018, Theorem 2) which we use to prove equation (4), which explicitly require these assumptions. I am not aware of any other such results in the literature which are sufficiently tight for the purposes of this proof. Secondly, our result relies on explicitly bounding $\sum_{j>d} \lambda_j$​ for which we use the eigenvalue decay assumption and the proof of the claim proved in Section D repeatedly employs Hoeffding's inequality for which we require the explicit eigenfunction growth assumption (for example, in line 590).
>
> The reviewer also mentions that they believe that assumption (R) does not seem natural. A similar comment was made by reviewer 2JjA, so I respond to this in the global response.
>
> I also respond to the reviewer's comment that the contributions are undetermined given previous work in the global response.
>
> In view of the importance of the entrywise error bound, fairness is one important reason and another is that such bounds can be used to show improved bounds for other algorithms which use these approximations (see response to Q5).
>
> I thank the reviewer for pointing out some minor typos, they will be corrected in the revision.
>
> In response to the reviewers questions:
>
> 1. Viewing the matrix $\tfrac{1}{n}K$ as a operator which acts on vectors $y\in\R^n$ (which we also view as a function from $\mathcal X$ to $\R$, associating each $i$ to $x_i$) with matrix multiplication, i.e.
>
>    $$
>    [\tfrac{1}{n}Ky]_i = \tfrac{1}{n}\sum_{j=1}^n K(i,j)y(j) = \tfrac{1}{n}\sum_{j=1}^n k(x_i,x_j)y(j)
>    $$
>
>     and taking the $n\to\infty$​ limit we have
>
>    $$
>    [\tfrac{1}{n}Ky]_i \to \int k(x_i,z) y(z) \:\text{d}\rho(z) = \mathcal K y.
>    $$
>
> 2. E.g. smooth radial kernels follow the eigenvalue decay assumption in (E) (see Belkin (2018)) regardless of the data generating measure. The exponential eigenfunction growth condition is then very mild.
>
> 3. I'm not aware of any lower bounds for the max-norm, although this would be an interesting direction for future research.
>
> 4. This optimisation problem is likely highly non-convex, but this would be an interesting avenue for future research.
>
> 5. Beyond fairness, for example, this max-norm bound could be used to show stability bounds for algorithms which use this low-rank approximation in place of the true kernel matrix. For example, Proposition 2 of Cortes et al. (2010) derives a stability bound for a SVM trained with a kernel approximation in terms of a spectral norm bound on the kernel approximation. For the values of $d$ allowed in our Theorem 1, this bound would not show consistency of the estimator learned with the kernel approximation, however a napkin calculation suggests that one could show consistency using our entrywise bound, although this is beyond the scope of this paper.
>
> *References:*
> - Mikhail Belkin. Approximation beats concentration? an approximation view on inference with smooth radial kernels. In Conference On Learning Theory, pages 1348–1361. PMLR, 2018.
> - Cortes, C., Mohri, M., & Talwalkar, A. (2010, March). On the impact of kernel approximation on learning accuracy. In *Proceedings of the thirteenth international conference on artificial intelligence and statistics* (pp. 113-120). JMLR Workshop and Conference Proceedings.

---

> > ### Comment · Reviewer_3p4Z · 2024-08-11
> > **Reponse**
> >
> > Thank you for your response. After the rebuttal, I go through the Appendix and believe the author has make great efforts in the proof, not as easy as we reviewers thought. But considering the fact that Lemma 1 should be corrected and the assumptions still lack some intuition (since it is hard to give some general examples), my score remain the same.

---

> > > ### Author Response · Authors · 2024-08-12
> > >
> > > I thank the reviewer for taking the time to go through the appendix, and I am happy to see their recognition of the complexity of some of the technical contributions there.
> > >
> > > I would argue that the correction to Lemma 1 is only minor, does not affect any other areas of the proofs and is now resolved.
> > >
> > > With regards to the difficultly in verifying the assumptions, this is a widespread limitation of *all* theoretical works on kernel methods which require knowledge of the spectral properties of the kernel, and I refer the reviewer to Section 2.2 of Barzilai and Shamir (2023) for an extended discussion of this point. Given this, I believe I take the best possible approach to making the assumptions interpretable to the reader. In Proposition 2, I show that in the special case of dot product kernels on the sphere (for which the spectral properties of the kernel *can* be easily computed), the assumptions can be replaced with a simple, highly-interpretable smoothness assumption on the kernel. I then show  experimentally that these results generalise to other data-generating measures using simulations and real datasets (using 4 Matérn kernels of differing smoothness). This is a standard approach, for example in the theoretical deep learning literature (e.g. Jacob et al. (2018), Bietti and Mairal (2019), Bietti and Bach (2020)), and I don't see that there is a better way of doing it.
> > >
> > >
> > >
> > > *References:*
> > >
> > > - Daniel Barzilai and Ohad Shamir. Generalization in kernel regression under realistic assumptions. *arXiv preprint arXiv:2312.15995, 2023*.
> > >
> > > - Bietti, A., & Bach, F. (2020). Deep equals shallow for ReLU networks in kernel regimes. *arXiv preprint arXiv:2009.14397*.
> > >
> > > - Bietti, A., & Mairal, J. (2019). On the inductive bias of neural tangent kernels. *Advances in Neural Information Processing Systems*, *32*
> > >
> > > - Jacot, A., Gabriel, F., & Hongler, C. (2018). Neural tangent kernel: Convergence and generalization in neural networks. *Advances in neural information processing systems*, *31*.

---

### Official Review · Reviewer_tJPJ · 2024-07-11

**Soundness:** 2
**Presentation:** 4
**Contribution:** 3
**Rating:** 5
**Confidence:** 4

**Summary:**

The authors consider the kernel matrices, formed by $n$ vectors i.i.d. drawn from a $p$-dimensional probability distribution $\rho$. Under several assumptions on the associated kernel operator on $L^2_{\rho}$, including the positive definiteness of the kernel and decay condition on the eigenvalues of the kernel, the authors prove an estimate on individual entries of the matrix kernel and those of the low-rank approximation of the kernel. Numerical experiments on the estimate error are done with both synthetic datasets and real-world datasets.

**Strengths:**

- The problem is a very fundamental one and it is considered both analytically and numerically.
- The writing is very clear and easy to read.

**Weaknesses:**

- Lemma 1 is wrong, and thus the proofs of the main results do not work.
Consider an extreme case where $a=0$ with probability $1$. Then, since $\pi$ is an orthogonal projection, $\| \pi_H(a) \| = 0$ and thus Lemma 1 fails. The main issue is that in the proof of Lemma 1, if $S_1 = \sum p_{ii} (\xi_i^2 - 1)$, then $E[S_1^2] = \sum_{i, j} p_{ii} p_{jj} E[\xi_i^2 - 1] E[\xi_j^2 - 1]$, which is different from $\sum_i p_{ii}^2 E[(\xi_i^2 - 1)^2]$ in (17), unless $E[\xi^2]=1$. As a result, (17) and the estimates on $P(E_+)$ and $P(E_-)$ fail.
-> The proofs of the main results would work after modifying Lemma 1 as suggested by the authors.

**Questions:**

- Is it possible to prove Lemma 1 with additional assumptions that are suitable to the current setting?
- In the proof of Lemma 1, there are other minor problems listed below.
1) In line 618, $\xi_i \in [0, 1]$ is wrong since the mean $\bar{x}$ is subtracted.
2) In the equation below line 621, why $\| \pi_H(x)\|^2 = \| \pi_H(\bar{x}) \|^2 + \| \pi_H(\xi) \|^2$?
3) In the equation below line 621 and several other places, $X$ should be $x$. Also, $\bar{\xi}$ should be $\xi$.

**Limitations:**

The work does not seem to have potential negative societal impact.

---

> ### Author Rebuttal · Authors · 2024-08-05
>
> I would like to start by thanking the reviewer for taking the time to work through my paper, in particular for working through the proofs in the appendix and for noticing a mistake in Lemma 1 which I address below. I was happy to read that they consider the problem to be very fundamental and that they found the writing to be very clear and easy to read.
>
> The reviewer is correct that there is indeed a mistake in Lemma 1, although this can be rectified with a simple modification of the lemma statement, which does not materially change the proof of the main result. As mentioned by the reviewer, in the proof of Lemma 1, I have implicitly assumed that the variance of each element of $a$ is one (which, in fact, by Popoviciu's inequality is not compatible with the assumption that each element of $a$ is uniformly bounded in $[0,1]$). This can be rectified by simply assuming that each element of $a$ has variance $\sigma^2$ and replacing each $q$ with $\sigma^2 q$ in the lemma. The lemma statement would then read:
>
> If $H$ is such that $\|\pi_H(\bar a)\| \leq 2(\sigma^2 q)^{1/4}$, then for any $t \geq 8$
>
> $$
> \mathbb{P} \left( \left| \|\pi_H(a)\| - \sigma q^{1/2} \right| \geq t \right) \leq 4 \exp(-t^2/32).
> $$
>
> Following this modification through in the proofs, $S_1$ defined in line 616 would become $S_1 = \sum_{i=1}^n p_{ii} (\xi_i^2 - \sigma^2)$ and therefore $S_1^2 = \sum_{i,j=1}^n  p_{ii} p_{jj}(\xi_i^2 - \sigma^2)(\xi_j^2 - \sigma^2)$. Now for each $i \in \{1,\ldots,n\}$, $(\xi_i^2 - \sigma^2)$ are independent mean zero random variables and therefore for $i \neq j$, we have that $\mathbb{E}\{(\xi_i^2 - \sigma^2)(\xi_j^2 - \sigma^2)\} = 0$. Therefore
>
> $$
> \mathbb{E}(S_1^2) = \sum_{i,j=1}^n  p_{ii} p_{jj}\mathbb{E}\{(\xi_i^2 - \sigma^2)(\xi_j^2 - \sigma^2)\} = \sum_{i=1}^n p_{ii}^2 \mathbb{E}\{ (\xi_i^2 - \sigma^2)^2 \}
> $$
>
> as required. From hereon, the proof of Lemma 1 follows through.
>
> In the main thread of the proof of Theorem 2 from line 206, we can set $\sigma^2$ equal to the variance of $k(x_1, y)$ where $y \sim \rho$. Provided that for all $x_1 \in \mathcal{X}$ we have that $\sigma^2$ is bounded away from zero, then the rest of the proof holds. I will add this non-degeneracy condition as an assumption to the theorem.
>
> Additionally, I thank the reviewer for pointing out two more typos. The equation below line 621 should contain an inequality $\leq$ rather than an equality. I will correct the typos relating to $x$ and $\xi$​.
>
> I will be interested to hear whether the reviewer is satisfied with this correction and if they are, their view of the paper as a whole.

---

> > ### Comment · Reviewer_tJPJ · 2024-08-10
> >
> > Thank you for your answers. It seems that the main result can be proved with the modified version of Lemma 1. (However, the modified version of Lemma 1 is less significant than the original one, since the new assumption $|\pi_H(\bar a)| \leq 2(\sigma^2 q)^{1/4}$ is stronger.) I will adjust my rating accordingly.

---

> > > ### Author Response · Authors · 2024-08-12
> > >
> > > I am glad to see that the reviewer is satisfied with my proposed modification to Lemma 1 and I thank them for updating their score. In response to their comment about the significance of the new lemma, I would argue that the new lemma is no less significant compared to the original one, especially in the asymptotic analysis considered in this paper.
> > >
> > > So far, the reviewer has only commented on Lemma 1 in the paper, which is only a small (but necessary) part of this work and is not the main contribution of the paper. I would be interested to know their opinion of the paper as a whole, now that the potential problem has been resolved.

---

### Author Rebuttal · Authors · 2024-08-05

I would like to start by thanking all the reviewers for taking the time to work through my paper and to write their reviews. I was happy to read that the reviewers consider the problem to be a very fundamental one, that the main results in the paper are new to the community, that the paper is clear, well-written and easy to follow, and that the experiments complement the theoretical results well.

I am grateful to reviewer tJPJ for pointing out a mistake in Lemma 1, which can be rectified with a simple modification of the lemma statement, and which does not materially change the proof of the main result. I outline this in the direct response to their review, and if they are satisfied with the change, I will be interested to hear their opinion of the paper overall.

Both reviewers 3p4Z and 2JjA mention that the assumptions (P) and (E) seem very natural and cover many common kernels, however both questioned how natural the assumption (R) is. In this global response, I will provide some high-level intuition as to where this assumption comes from, and why it is necessary. Reviewer 2JjA argues that Lemma 1 is only a slight generalisation of Lemma 68 in Tao and Vu (2011), and therefore does not represent a significant novel contribution, and reviewer 3p4Z argues that with previous works on random matrix theory, the proofs are straightforward. In this global response, I will emphasise why I do not believe this to be the case, and clarify where new technical insights were required to prove the results in the paper.

**On the assumption (R).** To employ Lemma 1, we require that the projection of a constant vector onto a subspace spanned by the eigenvectors of $K$, corresponding to a subset of the eigenvalues with index $i > d$, is sufficiently small. The natural population analog of this is to assume that the projection of a constant function onto the subspace spanned by the population eigenfunctions with index $i > j$, which I denote $\Gamma_j$, decays sufficiently quickly with $j$. This assumption aligns with the intuition that the frequency of each eigenfunction increases as its corresponding eigenvalue decreases. If the first eigenfunction is constant, as is the case in Special Case 2, all the remaining eigenfunctions are orthogonal to it, and so $\Gamma_j$ is zero whenever $j\geq 1$.  Reviewer 3p4Z asks if I can provide some examples where this isn't the case. Unfortunately, it is only possible to explicitly compute the eigenvalues and eigenfunctions of very special kernels with respect to very special measures. Typically, it is certain symmetries which allow them to be calculated, and in all the cases I am aware of, $\Gamma_j = 0$ for all $j\geq 1$. The assumption $a < b/16$ in (R) can be seen as a relaxation of this case, and the constant $16$ just pops out of what is possible with the proof technique I am using. I considered simply replacing (R) with the assumption that $\Gamma_j = 0$ for sufficiently large $j$, which is more restrictive but perhaps more natural seeming, however I preferred to stick with the slightly more involved and more general assumption.

**On the novelty of Lemma 1.** Reviewer 2JjA argues that Lemma 1 is only a slight generalisation of Lemma 68 in Tao and Vu (2011), and that since it follows a similar proof technique, does not represent a significant novel contribution. My Lemma 1 generalises Lemma 68 of Tao and Vu (2011) by allowing the mean $\bar a$ of the random vector $a$ to be non-zero. Establishing the condition on the relationship between subspace $H$ and the mean vector $\bar a$ for which such a concentration inequality can be derived was non-trivial and could not simply be guessed without working through the mathematical details. It is this innovation which represents the novelty of the lemma.

**On the novelty of the main proof.** Reviewer 3p4Z argues that with previous works on random matrix theory, the proofs of the main theorems are straightforward. I do not believe this to be the case and I will provide a summary of some of the technical challenges which arose in each part of this proof. Firstly, the core proof technique for the delocalisation bound (Theorem 2) follows a technique used in Tao and Vu (2011) which reduces the problem to (i) controlling the eigenvalues $\hat \lambda_i$ for $i$ in a constructed set $J$, and (ii) proving that the size of the orthogonal projection of a row of $K$ onto the subspace spanned by the eigenvectors $u_i$ for $i \in J$ is sufficiently large. The matrices considered in the paper of Tao and Vu (2011) and this paper are very different. Tao and Vu (2011) consider Wigner random matrices with independent mean-zero, unit variance entries, while we consider kernel matrices which have neither mean-zero nor independent entries, the randomness stemming from sampling the data points used to construct it. For this reason, (i) requires very different techniques - we use a concentration due to Valdivia (2018). (ii) also requires the new concentration inequality (Lemma 1) since our matrix does not have mean-zero entries. This new concentration inequality requires a condition on the mean of a row of the matrix and the aforementioned subspace. Proving that this condition holds in our setting (Section D) requires 7 pages of novel technical derivations. Additionally, understanding the necessary assumptions to make everything fit together is entirely non-trivial. For these reasons, I don't believe that this work can be reasonably described as "straightforward" application of previous work.

*References:*
- Terence Tao and Van Vu. Random matrices: Universality of local eigenvalue statistics. Acta Mathematica, 206(1):127 – 204, 2011.
- Ernesto Araya Valdivia. Relative concentration bounds for the spectrum of kernel matrices. arXiv preprint arXiv:1812.02108, 2018.

---

### Decision · Program_Chairs · 2024-09-25

**Decision:**

Accept (poster)

**Comment:**

The manuscript provides novel entrywise error bounty for low-rank approximations of kernel matrices. Of particular note, the arguments require (and the manuscript provides) a delocalization result for eigenvectors that may be of broader interest. While the delocalization result is stylistically similar to prior work, there are key differences in the regime in which it is valid. There is consensus that the results are interesting and technically correct (see below) and all of the reviewers are positive albeit weakly so. Moreover, some of the highlighted weaknesses are quite common for papers in this area—"unverifiable assumptions" are, unfortunately, common and not just a weakness of this particular work. Given the importance of the problem results of this type are valuable and the rebuttal effectively justified the assumptions to the extent possible.

Of note, the submitted version of the manuscript contained an error in Lemma 1, a reasonable fix was proposed during the rebuttal—it must be incorporated into the manuscript.

NB: while not particularly relevant to the overall rating of the manuscript, I will add that per my own read the "double descent" phenomena is alluded to in a rather ad hoc manner (admittedly the phrase has become somewhat ambiguous over time) and absent more clarification or study (which the authors suggest is out of scope) it may be best to omit it entirely.